# HyperNear: Unnoticeable Node Injection Attacks on Hypergraph Neural Networks

Tingyi Cai [1 2 3]   Yunliang Jiang [1 2 3 4]   Ming Li [5 1]   Lu Bai [6]   Changqin Huang [1]   Yi Wang [1]

## Abstract

With the growing adoption of Hypergraph Neural Networks (HNNs) to model higher-order relationships in complex data, concerns about their security and robustness have become increasingly important. However, current security research often overlooks the unique structural characteristics of hypergraph models when developing adversarial attack and defense strategies. To address this gap, we demonstrate that hypergraphs are particularly vulnerable to node injection attacks, which align closely with real-world applications. Through empirical analysis, we develop a relatively unnoticeable attack approach by monitoring changes in homophily and leveraging this self-regulating property to enhance stealth. Building on these insights, we introduce HyperNear, i.e., Node injEction Attacks on hypeRgraph neural networks, the first node injection attack framework specifically tailored for HNNs. HyperNear integrates homophily-preserving strategies to optimize both stealth and attack effectiveness. Extensive experiments show that HyperNear achieves excellent performance and generalization, marking the first comprehensive study of injection attacks on hypergraphs. Our code is available at https://github.com/ca1man-2022/HyperNear.

## 1. Introduction

Hypergraphs have emerged as a powerful framework for modeling higher-order relationships in complex systems, offering richer representations compared to traditional graphs. Their ability to capture intricate structures has enabled successful applications in domains like social networks and biological systems (Feng et al., 2024; Gao et al., 2024; Antelmi et al., 2023). However, the growing reliance on hypergraph-based models has raised concerns about their robustness and security (Hu et al., 2023; Chen et al., 2023).

Adversarial attacks, particularly those targeting Hypergraph Neural Networks (HNNs), pose serious risks to their reliability, potentially impacting sensitive areas such as healthcare and finance. While much research focuses on adversarial challenges in pairwise graphs (Zhang & Zitnik, 2020; Huang et al., 2017), the unique vulnerabilities of hypergraphs remain underexplored, highlighting the need for more resilient HNNs.

Among the various types of adversarial attacks, we focus on node injection attacks (Tao et al., 2021; Zou et al., 2021; Chen et al., 2022; Zhang et al., 2024). These attacks are stealthy and practical, as they avoid altering existing nodes or hyperedges. Figure 1(a) illustrates how injected nodes influence the network by embedding themselves within the hypergraph structure. However, we find that naive injection strategies often lead to significant topological changes, reducing both the effectiveness and stealthiness of the attack. Particularly, hypergraphs amplify minor perturbations due to their unique structure, making stealthiness a key challenge.

The most threatening adversarial strategies are those that achieve disruption while remaining imperceptible. To illustrate, consider a social network as shown in Figure 1(b), where a malicious actor aims to infiltrate without raising suspicion. The attacker strategically mimics the connections and attributes of legitimate users, effectively blending in through homophily (Bi et al., 2024; Wang et al., 2024; Luan et al., 2022), where similar nodes are more likely to connect. This principle of homophily provides a useful blueprint for adversarial unnoticeability, as it encourages alignment with the existing network structure. By aligning perturbations with homophily principles, we aim to design a framework

[1]Zhejiang Key Laboratory of Intelligent Education Technology and Application, Zhejiang Normal University, Jinhua, China [2]China-Mozambique Belt and Road Joint Laboratory on Smart Agriculture, Zhejiang Normal University, Jinhua, China [3]School of Computer Science and Technology, Zhejiang Normal University, Jinhua, China [4]School of Information Engineering, Huzhou University, Huzhou, China [5]Zhejiang Institute of Optoelectronics, Jinhua, China [6]Beijing Normal University, Beijing, China. Correspondence to: Yunliang Jiang <jyl2022@zjnu.cn>.

*Proceedings of the $42^{nd}$ International Conference on Machine Learning*, Vancouver, Canada. PMLR 267, 2025. Copyright 2025 by the author(s).

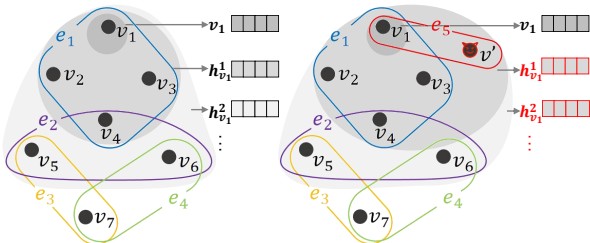

(a) Malicious impacts propagate along topology.

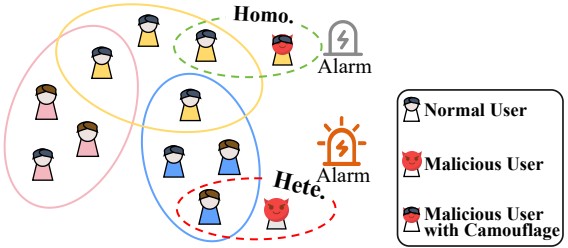

(b) Social networks subject to injection attacks: Homophily (Homo.) vs. Heterophily (Hete.) cases.

*Figure 1.* Schematic of hypergraph injection attacks.

that achieves both effectiveness and unnoticeability.

In this paper, we introduce HyperNear, the first node injection attack framework tailored for HNNs. We begin with a theoretical analysis to uncover HNNs vulnerabilities to injection attacks, highlighting the significant structural disruptions they cause. To ensure stealth, we conduct an empirical analysis revealing that naive attacks often drastically alter homophily, a key metric of node similarity, reducing their effectiveness. Based on these findings, we design HyperNear, incorporating homophily constraints to achieve imperceptible perturbations. Extensive experiments on five real-world datasets demonstrate HyperNear's effectiveness, generalization, and stealth, outperforming baseline methods.

To summarize, our paper makes the following contributions:

- **From the perspective of problem definition**, we make the first attempt to study the research on injection attacks on hypergraphs to extend the existing research on hypergraph applications to consider security and to advance the practical applications of hypergraph systems.

- **From the perspective of theoretical insights**, we address the specific challenges posed by adversarial attacks on hypergraphs by providing a theoretical analysis that demonstrates the susceptibility of hypergraph models to our proposed attack strategies. This analysis further informs and guides the design of our algorithm for adversarial attacks on hypergraphs.

- **From the perspective of algorithmic development**,

we propose a new approach termed HyperNear, which provides a stealth-enhanced hypergraph injection attack strategy by utilizing the homophily metric as a constraint.

- **Our HyperNear shows superior attack performance** and stealth through extensive experiments. Moreover, it demonstrates good transferability across different backbone hypergraph networks.

## 2. Preliminaries and Problem Definition

### 2.1. HNNs

Given a hypergraph $\mathcal{G} = (\mathcal{V}, \mathcal{E}, \mathbf{W})$, which consists of a set of nodes $\mathcal{V}$ and a set of hyperedges $\mathcal{E}$. For each node $v$, we denote its class by $y_v$, and the structure of a hypergraph is usually represented by an incidence matrix $\mathbf{H} \in \mathbb{R}^{|\mathcal{V}| \times |\mathcal{E}|}$. Each entry $\mathbf{H}(v, e)$ denotes whether a node $v$ is in the hyperedge $e$ or not, i.e.,

$$\mathbf{H}(v, e) = \begin{cases} 1 & \text{if } v \in e \\ 0 & \text{if } v \notin e \end{cases}. \quad (1)$$

Then, we can define the degree of a node $v \in \mathcal{V}$ and the average degree of hyperedge $e \in \mathcal{E}$ as follows:

$$d_v = \sum_{v \in \mathcal{V}} \mathbf{H}(v, e), \quad d_e = \frac{1}{|e|} \sum_{v \in e} d_v. \quad (2)$$

For clarity, Table 6 in the Appendix A summarizes the symbols and definitions used in this paper.

### 2.2. Homophily

Homophily in hypergraphs refers to the tendency of nodes with similar characteristics to form higher-order connections within a hyperedge (Li et al., 2025). We define homophily for hypergraphs based on the similarity of node labels within each hyperedge. Specifically, the homophily rate for hyperedges is calculated as follows:

$$\mathcal{H} = \frac{1}{|\mathcal{E}|} \sum_{m=1}^{|\mathcal{E}|} \frac{|\{u, v\} : \{(u, v) \in e_m \wedge (y_u = y_v)\}|}{C_{v_m}^2}. \quad (3)$$

We call this the node label-based homophily rate, where $e_m$ is the $m$-th hyperedge and $C_{v_m}^2$ is the combination number, representing the number of possible ways to choose two nodes from $v_m$ nodes.

### 2.3. Problem Definition

We focus on node injection attacks, a stealthy adversarial approach that introduces new nodes into the hypergraph without altering the original structure. These attacks are subtle but can severely distort model predictions. In real-world applications, they enable adversarial manipulation of

the hypergraph at minimal cost, posing a serious threat. We define and formalize the objectives of hypergraph injection attacks.

**Definition 2.1.** (Hypergraph Injection Attack) *Given a hypergraph* $\mathcal{G} = (\mathcal{V}, \mathcal{E}, \mathbf{W})$ *with an incidence matrix* $\mathbf{H} \in \mathbb{R}^{|\mathcal{V}| \times |\mathcal{E}|}$ *and a node feature matrix* $\mathbf{X} \in \mathbb{R}^{|\mathcal{V}| \times d}$, *a hypergraph injection attack introduces a set of new nodes* $\widehat{\mathcal{V}}$ *and new hyperedges* $\widehat{\mathcal{E}}$ *into the hypergraph, while leaving the edges and features of the original nodes* $\mathcal{V}$ *unchanged. Formally, the attack constructs a modified hypergraph* $\mathcal{G}'$ *with the following incidence matrix:*

$$\mathbf{H}' = \begin{bmatrix} \mathbf{H} & \mathbf{I}_e \\ \mathbf{I}_v & \widehat{\mathbf{H}} \end{bmatrix}, \quad (4)$$

*where* $\mathbf{I}_e \in \mathbb{R}^{|\mathcal{V}| \times |\widehat{\mathcal{E}}|}$ *represents the connections between original nodes and new hyperedges,* $\mathbf{I}_v \in \mathbb{R}^{|\widehat{\mathcal{V}}| \times |\mathcal{E}|}$ *represents the connections between new nodes and original hyperedges, and* $\widehat{\mathbf{H}} \in \mathbb{R}^{|\widehat{\mathcal{V}}| \times |\widehat{\mathcal{E}}|}$ *represents the connections between new nodes and new hyperedges.*

**Definition 2.2.** (Optimization Objective for Hypergraph Injection Attacks) *The goal of the hypergraph injection attack is to introduce minimal perturbations to the hypergraph* $\mathcal{G}$ *in the form of additional incidence matrices* $\mathbf{H}'$ *and node feature matrices* $\mathbf{X}'$, *while minimizing the impact on model performance. Formally, the optimization problem is defined as:*

$$\max_{\mathbf{H}', \mathbf{X}'} \mathcal{L}\left(f_{\theta^*}(\mathbf{H}', \mathbf{X}')\right) \text{ s.t. } \theta^* = \arg\min_{\theta} \mathcal{L}\left(f_\theta(\mathbf{H}, \mathbf{X})\right), \quad (5)$$

*where* $f_\theta(\cdot)$ *is the model parameterized by* $\theta$, $\mathcal{L}(\cdot)$ *is the loss function (such as cross entropy). The objective is to maximize the model's loss on the adversarial hypergraph* $\mathcal{G}'$ *while ensuring that the original model's parameters* $\theta^*$ *are optimized to minimize the loss on the unperturbed hypergraph.*

## 3. Impact of Attacks on Topology and Homophily of Hypergraphs

In this section, we investigate the impact of adversarial attacks on HNNs, presenting two key findings.

- **Finding 1: Hypergraph topological vulnerability.** We demonstrate that hypergraphs are highly vulnerable to adversarial attacks, where even small perturbations may produce amplified effects due to the intricate interdependencies within hyperedges. This amplification poses unique challenges for developing attacks that are both effective and unnoticeable, as minor structural changes can propagate unpredictably.

- **Finding 2: Naive attacks disrupt homophily.** We observe that naive attacks, which are not designed with

hypergraph structures in mind, tend to significantly disrupt homophily, a metric that reflects the similarity between connected nodes. This sensitivity motivates using homophily to quantify subtle structural changes and refine attack strategies.

### 3.1. Topological Vulnerability of Hypergraphs

We express the general hypergraph convolution process (Chien et al., 2022) as an aggregation process at node $v$:

$$\mathbf{h}_v^{(k)} = \phi(\mathbf{h}_v^{(k-1)}, f(\{\mathbf{h}_u^{(k-1)} \mid u \in e_j, e_j \in \mathcal{R}_v\})), \quad (6)$$

where $\mathcal{R}_v = \cup_{e_j \in \mathcal{E}} \{e_j \mid v \in e_j\}$ is the set of all hyperedges containing node $v$, and $e_j (j = 1, 2, \ldots, |\mathcal{E}|)$ denotes the $j$-th hyperedge. The functions $\phi(\cdot)$ and $f(\cdot)$ are vector-valued functions that update the node features based on the information from neighboring nodes.

In adversarial settings, we focus on the hypergraph injection attack discussed in Sec. 2.3, which allows attackers to target specific regions of the hypergraph with limited access to the full network. To model the impact of such attacks on the hypergraph structure, we consider perturbations through node injections, leading to changes in both topology and feature representations.

**Theorem 3.1** (Propagation of Perturbation in HNNs). *Given a hypergraph with node* $v$ *and its feature vector* $\mathbf{h}_v^{(k)}$ *updated by the aggregation function* $\phi$ *as defined in Eq. (6), a perturbation* $\Delta \mathcal{R}_v$ *in the topology results in a feature change* $\Delta \mathbf{h}_v^{(k)}$ *that satisfies:*

$$\Delta \mathbf{h}_v^{(k)} = \underbrace{\sum_{t=0}^{n} \frac{\partial \phi}{\partial f_t} \Delta f_t}_{\text{Sensitivity of } \phi \text{ to t-hop features}} + \underbrace{O((\Delta f_t)^2)}_{\substack{\text{Nonlinear higher-order} \\ \text{effects significant} \\ \text{under large perturbations}}},$$
$$(7)$$

*where* $\Delta f_t = f_t' - f_t$ *represents the change in the* $t$-hop *aggregation function. This propagation effect highlights the sensitivity of hypergraph-based models to structural perturbations.*

*Proof.* Using Eq. (6), the feature vector $\mathbf{h}_v^{(k)}$ is updated by aggregating features from neighbors across $t$-hop distances. When a perturbation $\Delta \mathcal{R}_v$ alters the hypergraph structure, it modifies the aggregation function $f_t(\cdot)$ to $f_t'(\cdot)$. A first-order Taylor expansion of $\phi$ yields:

$$\mathbf{h}_v'^{(k)} = \mathbf{h}_v^{(k)} + \sum_{t=0}^{n} \frac{\partial \phi}{\partial f_t} \Delta f_t + O((\Delta f_t)^2), \quad (8)$$

where $\Delta f_t$ represents the perturbation. Subtracting $\mathbf{h}_v^{(k)}$ from $\mathbf{h}_v'^{(k)}$ gives the desired result. $\square$

**Corollary 3.2** (Amplification Effect of Aggregation). *For a weighted aggregation method, the perturbation $\Delta f_t$ can be expressed as:*

$$\Delta f_t = \underbrace{\sum_{\{u \in e_j | e_j \in \mathcal{R}_v\}} \left( \Delta w_u^t \mathbf{h}_u^{(k-1)} \right)}_{\text{Weight-induced perturbation}}$$

$$+ \underbrace{\sum_{\{u \in e_j | e_j \in \mathcal{R}_v\}} \left( w_u^t \Delta \mathbf{h}_u^{(k-1)} \right)}_{\text{Feature-induced perturbation}}, \quad (9)$$

*where $\Delta w_u^t$ and $\Delta \mathbf{h}_u^{(k-1)}$ represent the changes in the hyperedge weights and the node features, respectively. This result indicates that changes in hyperedge weights $\Delta w_u^t$ or node features $\Delta \mathbf{h}_u^{(k-1)}$ can amplify the overall perturbation. In particular, when certain hyperedges have larger weights or exhibit more significant feature changes, they disproportionately influence the overall perturbation, causing a more substantial impact on the node feature updates.*

**Remarks.** Figure 10 illustrates the sensitivity of node $v$ to $t$-hop neighbors and the nonlinear higher-order effects in HNNs. More detail is shown in the Appendix B.

Through this analysis, it becomes evident that hypergraph topology is particularly vulnerable to small, targeted perturbations. A single perturbation can propagate through the structure, affecting multiple nodes simultaneously and highlighting the fragility of hypergraphs. However, introducing new nodes and hyperedges often leads to noticeable changes in the overall topology, making it easier for defenders to detect potential breaches. The key question is ***how can we design more subtle and unnoticeable attacks that exploit these topological vulnerabilities in hypergraphs?***

### 3.2. Significant Shifts in Homophily Caused by Naive Attacks

Building on the vulnerability analysis from Sec. 3.1, we focus on the impact of adversarial attacks on homophily, a key structural property of hypergraphs. Specifically, our empirical studies show a significant reduction in the homophily ratio after attack.

**Proposition 3.3.** *Naive adversarial attacks cause a significant reduction in the homophily ratio of hypergraphs, which negatively impacts the network's ability to preserve structural relationships between nodes. This reduction in homophily leads to a decrease in the predictive accuracy of Hypergraph Neural Networks (HNNs).*

**Observation 1.** *The experimental results presented in Table 1 show a significant decrease in the homophily ratio after adversarial attacks. This reduction occurs even when the attacks are not specifically designed to target structural*

Table 1. Homophily ratio changes in the hypergraph neural network (UniSAGE) after adversarial attack.

| DATASETS | CORA-CA | DBLP-CA | CITESEER | CORA | PUBMED |
|---|---|---|---|---|---|
| BEFORE | 0.6129 | 0.6381 | 0.7104 | 0.7079 | 0.7710 |
| AFTER | 0.5416↓ | 0.5508↓ | 0.5859↓ | 0.5915↓ | 0.5888↓ |

*vulnerabilities, demonstrating that simple perturbations can cause noticeable changes in hypergraph structure.*

These results align with expectations (Zhu et al., 2022; Chen et al., 2022), as homophily, which reflects the similarity of nodes within the same hyperedge, plays a key role in HNNs' predictive accuracy. Our observation shows that naive attacks disrupt homophily patterns, suggesting its potential as a metric for assessing attack stealth. Minimizing homophily changes enables more covert attacks that effectively disrupt hypergraph structures, while also providing a basis for monitoring and refining attack designs.

While attackers typically lack complete node labels, consistent with real-world attack scenarios, we propose the concept of node feature homophily (Wang et al., 2025), which quantifies node similarity based on the aggregation of features from connected nodes.

**Definition 3.4.** (Feature-based Homophily in Hypergraphs) *The homophily of a hypernode $v$ can be defined as the similarity between the features of hypernode $v$ and the aggregated features of its neighboring nodes:*

$$FHH = \text{sim}(\mathbf{H}_v, \mathbf{X}_v),$$
$$\mathbf{H}_v = \sum_{e \in \mathcal{R}_v} \frac{1}{\sqrt{d_e} \cdot \sqrt{d_v}} \phi(\{\mathbf{X}_j\}_{j \in e}), \quad (10)$$

*where $d_e = \frac{1}{|e|} \sum_{i \in e} d_i$ is the average degree of hyperedge $e$ and $sim(\cdot)$ is a similarity metric, such as cosine similarity. The $\mathcal{R}_v$ denotes the set of all hyperedges containing node $v$. In this paper, homophily ratio is measured using the $FHH$.*

**Discussion.** Our analysis shows that adversarial attacks severely disrupt hypergraph structures, with a significant reduction in homophily observed in our experiments. This highlights that successful attacks not only destabilize the structure but also leave detectable traces in homophily patterns. Since homophily is critical for HNNs performance, we propose using it as a metric to design stealthier attacks. Minimizing disruptions to homophily can make interventions less noticeable while still effectively degrading network performance.

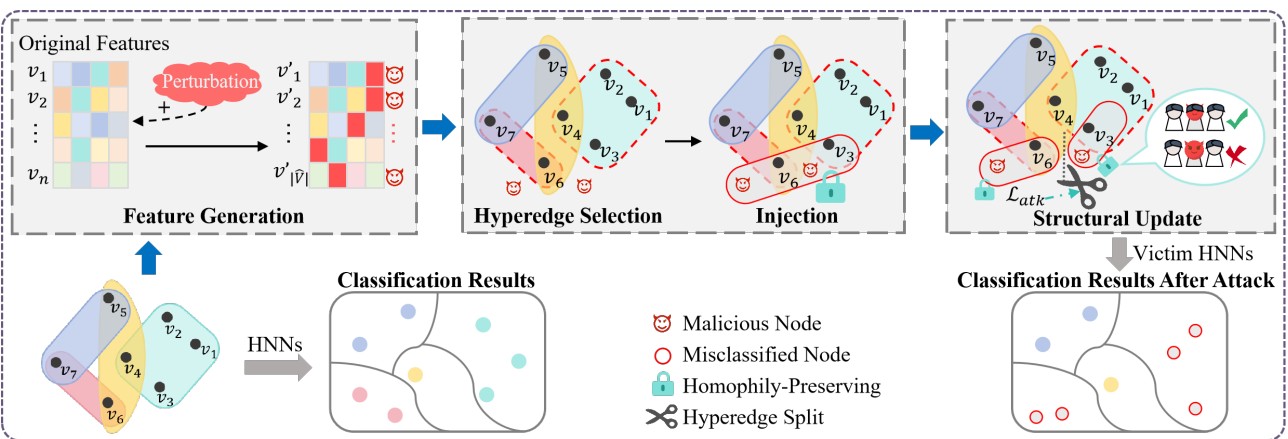

*Figure 2.* Systematic framework of HyperNear and corresponding evaluations. The process includes: (1) Feature generation to create adversarial node features, (2) Hyperedge selection to determine the target hyperedges for injection, (3) Injection of adversarial nodes into the hypergraph, and (4) Structural update to refine hyperedge connections, ensuring the attack remains effective and unnoticeable.

## 4. Methodology

In this section, we present HyperNear, i.e., Node injEction Attacks on hypeRgraph neural networks, the first node injection attack framework tailored for HNNs. The overall process of HyperNear is illustrated in Figure 2. We begin by introducing the design idea of the general framework for hypergraph injection attack, followed by the integration of homophily-preserving adversarial objectives to enhance the effectiveness of the attacks.

### 4.1. The HyperNear Framework

The framework comprises four core steps: feature generation, hyperedge selection, injection, and structural update. This structured approach addresses the unique challenges of hypergraph manipulation and provides a foundation for future adversarial attack research on hypergraphs.

**Feature Generation.** The first step in our attack is generating features for the injected nodes. To make them indistinguishable from original nodes, we adopt a perturbation-based approach. Specifically, the feature vector of the injected node is created by adding controlled perturbations to the existing node features. This ensures that the injected nodes share characteristics with the original graph (Bai et al., 2022), while allowing for the necessary deviations to achieve adversarial effects. Formally, for each injected node, we denote its feature vector as $\mathbf{x}_{inj}$, generated from an original node feature $\mathbf{x}_{ori}$ as:

$$\mathbf{x}_{inj} = \mathbf{x}_{ori} + \eta, \qquad (11)$$

where $\eta$ is a small perturbation to the feature space.

**Hyperedge Selection.** In our framework, we use a randomized selection strategy to choose the hyperedges of the injected nodes. Although random, this selection process has two key advantages: it avoids bias in the attack design, making detection harder, and it ensures flexibility in a variety of hypergraph topologies.

**Injection.** Subsequently, we adopt a fully connected injection strategy as the initialized injection state, i.e., all injected nodes are connected to nodes in the selected hyperedge. This approach extends the influence of the injecting node and maximizes its potential to disrupt the network's predictive capabilities. This approach amplifies the influence of the injected node, maximizing its potential for disrupting the network's prediction capabilities.

**Structural Update.** To increase the adversarial impact, we split the injected hyperedge that is fully connected based on the gradient information obtained during backpropagation. Updates are guided based on the principle of enlarging the classification loss, adjusting the connections within the hyperedge to ensure that perturbations caused by the injected nodes maximize the damage to the model performance. This process can be represented as

$$S_{inj} = \arg\max_{S} \mathcal{L}_{cls}(f(\mathbf{H}_{update}), y), \qquad (12)$$

where $S_{inj}$ represents the splitting strategy applied to the injected hyperedge, and $\mathbf{H}_{update}$ denotes the hypergraph structure after a random split of the injected hyperedge. The goal is to find a split $S$ that maximizes the classification loss $\mathcal{L}$, thereby increasing the adversarial influence of the attack. Our analysis shows that hypergraphs are susceptible to both feature-based attacks and structural attacks, which together can degrade the performance of hypergraph-based models drastically. Our attack strategy achieves dual perturbations of features and structure, thereby enhancing the effectiveness of the attack.

### 4.2. Homophily-Preserving Adversarial Objective

Maintaining homophily is critical for designing unnoticeable attacks. Our empirical findings reveal that adversarial attacks significantly disrupt hypergraph homophily, which measures the similarity between a node's features and its neighbors. Disrupting homophily compromises structural coherence, making attacks more detectable.

To address this, we propose a homophily-preserving adversarial objective with the following constraints:

$$\mathcal{L}_{FHH} = \frac{(FHH_{after} - FHH_{before})^2}{\tau}, \qquad (13)$$

where $FHH_{before}$ and $FHH_{after}$ are the homophily ratios before and after the attack, and $\tau$ controls the allowable change.

This objective is integrated into the overall adversarial loss as:

$$\max_{||\mathcal{G}'-\mathcal{G}|| \leq \mathcal{B}} \mathcal{L}_{atk} = \mathcal{L}_{cls} - \lambda \mathcal{L}_{FHH}, \qquad (14)$$

where $\lambda$ balances attack effectiveness and homophily preservation, and $\mathcal{B}$ is the allowed perturbation budget. Further details, including the derivation of homophily perturbations, can be found in Appendix A.1 .

In summary, our approach introduces the first adversarial attack on hypergraphs with homophily constraints, achieving effective yet unnoticeable attacks. Algorithm 1 details the implementation. Our method aligns with three attack types: *poisoning*, *global*, and *black-box*. Poisoning attacks alter training data, global attacks degrade performance with minimal cost, and black-box setups simulate real-world conditions with limited information. Addressing node injection in these contexts is crucial for enhancing HNNs robustness.

## 5. Experiments

In this section, we present experiments to evaluate the effectiveness of the proposed HyperNear, addressing the following research questions:

**RQ1:** How effective is HyperNear compared to state-of-the-art attack methods?

**RQ2:** How transferable is HyperNear across different hypergraph architectures?

**RQ3:** How stealthy is HyperNear?

**RQ4:** How do hyperparameters affect HyperNear's performance?

### 5.1. Experimental Setups

**Datasets.** We evaluate our models on five real-world hypergraph datasets for hypernode classification tasks, including DBLP (Rossi & Ahmed, 2015), Pubmed, Citeseer and Cora (Sen et al., 2008). We use the same preprocessed hypergraphs as those provided in the official implementations of HyperGCN (Yadati et al., 2019) and UniGNN (Huang & Yang, 2021), ensuring that attack effects can be compared. More detail is in Appendix A.4.

**Baselines.** Since ours is the first work to perform a global adversarial attack on HNNs in a black-box setting, we compare with methods adapted from graph theories. It is briefly described as follows:

- Random Attack (Random): We believe that the method of randomized attacks can effectively illustrate the importance of understanding hypergraph structures. To ensure a fair comparison, we utilize a random injection attack as a benchmark for randomization, maintaining a certain level of concealment. This involves randomly generating node features and then selecting nodes for one-to-one injection.

- Node Degree Attack (NDA) (Zhang et al., 2024): Node degree is a significant metric for evaluating graph structure, and previous studies have demonstrated that attacking nodes with lower degrees can impair graph performance. Therefore, we extend the concept to hypergraphs by proposing a NDA method, which targets nodes with the smallest degrees and modifies their features.

- Fast Gradient Attack (FGA) (Chen et al., 2018): Gradient attacks are also a classic attack strategy. We extend this approach to hypergraphs by targeting the features of hypernodes with the highest absolute gradient values in the hypergraph proxy model during each attack iteration.

**Experimental Details.** Our experiments adopt a strict black-box setup, restricting any model querying and granting access only to the incidence and feature matrices of the input data. All methods are evaluated under identical budget constraints. The victim and target models are assumed identical, with the victim model being the pre-trained agent used in the attacks. More detail is provided in the Appendix A.2.

### 5.2. Performance Comparison (RQ1)

To answer **RQ1**, we compare HyperNear with three baselines on four victim HNNs in a black-box setting, as shown in Table 2. HyperNear demonstrates superior attack effectiveness in most cases, significantly reducing the classification accuracy of the victim models across five datasets. This indicates that HyperNear effectively exploits the vulnerabilities of HNNs in generating adversarial perturbations. The poor performance of the Random attack validates our intuition that injection attacks, while stealthier, require

*Table 2.* Comparison of classification accuracy (%) of several attack models. The classification performance of the hypergraph network is used as a measure of the effectiveness of the global attack, the lower the better, with the best results highlighted in bold and shaded in gray, and sub-optimal results shaded in blue. Red represents the invalid result of random injection attack.

| VICTIM MODEL | METHODS | CORA-CA | DBLP-CA | CITESEER | CORA | PUBMED |
|---|---|---|---|---|---|---|
| UNISAGE | CLEAN | 74.13 ± 1.23 | 89.01 ± 0.11 | 61.72 ± 1.84 | 67.44 ± 2.09 | 74.31 ± 1.61 |
| | RANDOM | 73.86 ± 1.20 | 89.00 ± 0.11 | 61.23 ± 1.64 | 67.38 ± 2.19 | 74.49 ± 1.55 |
| | NDA | 71.70 ± 1.07 | 84.93 ± 0.15 | 59.55 ± 1.73 | 64.96 ± 2.03 | 72.50 ± 1.28 |
| | FGA | 71.58 ± 1.95 | 88.57 ± 0.22 | 58.69 ± 2.02 | 63.57 ± 2.01 | 71.57 ± 3.12 |
| | HYPERNEAR (OURS) | **60.24 ± 2.10** | **80.12 ± 4.29** | **47.47 ± 4.80** | **49.35 ± 7.41** | **55.31 ± 8.13** |
| UNIGIN | CLEAN | 74.45 ± 1.12 | 89.14 ± 0.11 | 61.85 ± 1.87 | 67.69 ± 2.09 | 74.87 ± 1.50 |
| | RANDOM | 74.16 ± 1.25 | 89.13 ± 0.14 | 61.15 ± 1.61 | 67.49 ± 2.26 | 75.07 ± 1.42 |
| | NDA | 71.98 ± 1.09 | 84.92 ± 0.13 | 59.70 ± 1.79 | 64.98 ± 2.17 | 72.65 ± 1.41 |
| | FGA | 71.79 ± 1.58 | 88.58 ± 0.24 | 57.35 ± 2.23 | 63.48 ± 2.24 | 72.17 ± 2.42 |
| | HYPERNEAR (OURS) | **65.86 ± 1.47** | **80.16 ± 4.26** | **47.47 ± 4.80** | **49.35 ± 7.43** | **55.31 ± 8.12** |
| UNIGCN | CLEAN | 75.74 ± 0.88 | 88.83 ± 0.16 | 64.29 ± 1.16 | 70.53 ± 1.22 | 75.53 ± 1.00 |
| | RANDOM | 76.36 ± 1.27 | 88.82 ± 0.16 | 64.47 ± 1.12 | 70.77 ± 1.09 | 75.41 ± 1.07 |
| | NDA | 73.59 ± 0.74 | **84.60 ± 0.13** | 62.09 ± 1.44 | 68.27 ± 0.98 | 73.12 ± 0.89 |
| | FGA | 75.16 ± 0.93 | 88.80 ± 0.19 | 63.28 ± 1.40 | 69.60 ± 1.11 | 74.81 ± 1.15 |
| | HYPERNEAR (OURS) | **67.55 ± 0.67** | 85.45 ± 0.48 | **56.47 ± 0.75** | **56.69 ± 1.76** | **68.13 ± 0.87** |
| UNIGAT | CLEAN | 76.29 ± 1.22 | 88.85 ± 0.11 | 64.95 ± 1.40 | 70.51 ± 1.49 | 75.92 ± 0.84 |
| | RANDOM | 76.42 ± 1.10 | 88.83 ± 0.11 | 64.62 ± 0.93 | 71.43 ± 1.60 | 75.70 ± 1.17 |
| | NDA | 73.61 ± 1.31 | 84.74 ± 0.14 | 62.23 ± 0.94 | 68.16 ± 1.50 | 73.42 ± 0.87 |
| | FGA | 74.97 ± 1.35 | 88.69 ± 0.19 | 63.40 ± 1.34 | 69.53 ± 1.38 | 74.87 ± 1.06 |
| | HYPERNEAR (OURS) | **67.50 ± 2.30** | **84.36 ± 0.16** | **56.52 ± 0.59** | **57.13 ± 1.92** | **68.38 ± 1.18** |

thoughtful design for hypergraph models. Randomly injected nodes sometimes even improved model performance, as highlighted in red in Table 2. Although NDA and FGA are more reasonable alternatives, they are designed for traditional graphs and fail to fully leverage hypergraph-specific properties. In contrast, HyperNear, tailored for hypergraphs, disrupts the model's learning process more effectively. Notably, HyperNear achieves consistent performance across different datasets and victim models, highlighting its effectiveness and generalization capability.

### 5.3. Transferability Analysis (RQ2)

In addition to the four models demonstrated in Table 2, we also evaluate the transferability of HyperNear in four other classical HNN models, including HyperGCN (Yadati et al., 2019), HGNN (Feng et al., 2019), ED-HNN (Wang et al., 2023), and UniGCNII (Huang & Yang, 2021). Transferability is critical to understanding whether HyperNear can effectively compromise different hypergraph models in black-box settings, demonstrating its generality across various backbones. The average classification results over 10 runs, both before and after the attack, are presented in Table 3, demonstrating that HyperNear maintains consistent aggressiveness across all tested architectures. Such high transferability is advantageous for real-world scenarios

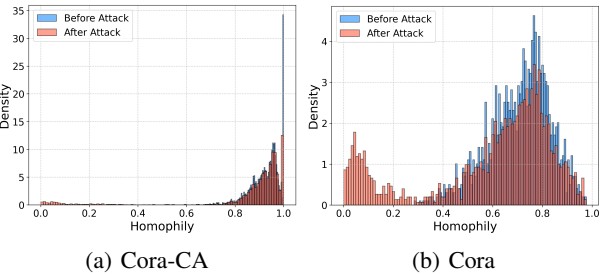

(a) Cora-CA      (b) Cora

*Figure 3.* Changes in the homophily ratio. Left: UniGCN is trained on the Cora-CA dataset. Right: UniSAGE is trained on the Cora dataset.

where attackers may not have detailed knowledge of the target model architecture.

### 5.4. Visualization (RQ3)

To answer **RQ3**, we analyze the homophily ratio and data distribution following the attack. As shown in Figure 3, the homophily ratio remains largely unchanged, indicating that HyperNear introduces perturbations in a manner that preserves the overall structural properties of the hypergraph. Despite this, it effectively degrades the predictive performance of the target model.

*Table 3.* Performance comparison of HyperNear on different HNNs.

| VICTIM MODEL | CORA | | DBLP-CA | | PUBMED | |
|---|---|---|---|---|---|---|
| | BEFORE | AFTER | BEFORE | AFTER | BEFORE | AFTER |
| HYPERGCN | $73.25 \pm 2.77$ | $49.56 \pm 5.91\downarrow$ | $87.80 \pm 0.41$ | $76.04 \pm 3.02\downarrow$ | $78.84 \pm 2.59$ | $64.06 \pm 5.71\downarrow$ |
| HGNN | $79.01 \pm 0.97$ | $52.19 \pm 1.58\downarrow$ | $91.01 \pm 0.22$ | $73.62 \pm 0.39\downarrow$ | $83.23 \pm 0.35$ | $67.18 \pm 0.61\downarrow$ |
| ED-HNN | $81.21 \pm 1.00$ | $62.11 \pm 1.88\downarrow$ | $91.93 \pm 0.20$ | $83.27 \pm 0.32\downarrow$ | $88.32 \pm 0.42$ | $78.42 \pm 0.47\downarrow$ |
| UNIGCNII | $77.52 \pm 1.12$ | $69.44 \pm 1.38\downarrow$ | $90.30 \pm 0.24$ | $86.25 \pm 0.34\downarrow$ | $86.08 \pm 0.48$ | $81.42 \pm 0.57\downarrow$ |

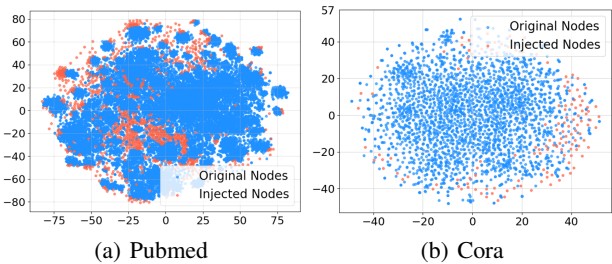

(a) Pubmed        (b) Cora

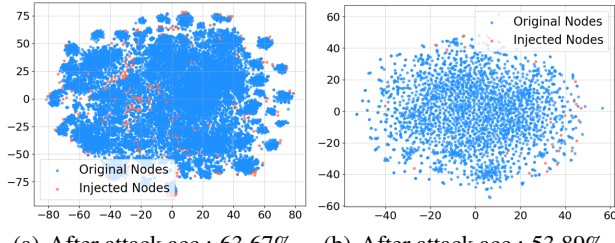

(a) After attack acc.: 63.67%.  (b) After attack acc.: 53.89%.

*Figure 4.* T-SNE visualization of data distribution after attack. The victim model is UniGAT.

*Figure 6.* Data distribution after HyperNear when $\alpha = 1\%$. Left: UniGAT is trained on the Pubmed dataset. Right: UniGAT is trained on the Cora dataset.

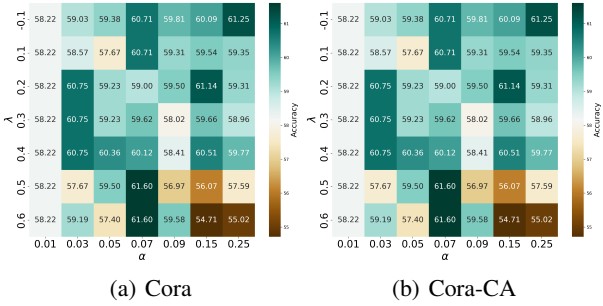

(a) Cora        (b) Cora-CA

*Figure 5.* Heatmap of classification accuracy after HyperNear in the $(\lambda, \alpha)$ parameter space.

We further visualize the data distribution after the attack using t-SNE (Hinton & Van Der Maaten, 2008) in Figure 4, which demonstrating that HyperNear does not drastically alter the overall embedding distribution in most cases. This effect is especially evident in Cora-CA (Figure 4(a)), where the distribution remains largely indistinguishable from the original, owing to its more structured hypergraph construction.

These analyses collectively demonstrate that HyperNear achieves relatively unnoticeable adversarial behavior across multiple datasets, with its detectability varying depending on the underlying hypergraph structure.

**Remarks.** The unnoticeability of HyperNear fundamentally stems from its homophily-aware perturbation strategy,

wherein the injected nodes are crafted to align with existing structural patterns within the hypergraph. This design enables the attack to evade detection while preserving key topological properties. While the empirical analyses above strongly suggest that HyperNear achieves improved unnoticeability, we acknowledge that a formal theoretical characterization of detectability under hypergraph-specific metrics remains an open challenge. Addressing this would further solidify the unnoticeability claim.

In addition, we note that most existing adversarial defenses for graphs may not transfer directly to hypergraph settings, owing to fundamental differences in representation and connectivity. As such, designing specialized defense mechanisms tailored for hypergraph neural networks is an important and promising direction for future research.

### 5.5. Hyperparameter Study (RQ4)

We analyze the impact of key parameters in HyperNear, focusing on the homophily preservation parameter $\lambda$ and the injection rate $\alpha$, which controls the number of injected hypernodes. To evaluate the sensitivity of HyperNear's effectiveness, we measure test accuracy across a parameter grid. Specifically, we vary $\lambda$ over $\{-0.1, 0.1, 0.2, \ldots, 0.6\}$ and $\alpha$ over $\{0.01, 0.03, \ldots, 0.09, 0.15, 0.25\}$. Figure 5 presents heatmaps for UniSAGE (the victim model), showing performance sensitivity across this parameter space. These results demonstrate that appropriate tuning of $\lambda$ and $\alpha$ enables HyperNear to balance attack effectiveness and stealth.

To further illustrate HyperNear's impact, we visualize the node embedding distribution post-attack. Figure 6 (setup consistent with Figure 4) highlights that reducing the injection rate ($\alpha = 1\%$) enhances the unnoticeability of the attack while maintaining a significant degradation in model performance.

## 6. Related Works

Adversarial attacks on graph data (Zügner et al., 2020; Gosch et al., 2023) are mainly categorized into modification (Zügner et al., 2018; Zügner & Günnemann, 2019) and injection attacks (Zou et al., 2021; Fang et al., 2024). While much of the research focuses on traditional graphs (Wu et al., 2019), the security of hypergraphs, which capture higher-order relationships, has been less explored. HNNs (Kim et al., 2024; Gao et al., 2020) are powerful tools for complex data (La Gatta et al., 2022; Shao et al., 2020; Yang et al., 2019), but their vulnerability to adversarial attacks remains understudied. Previous work (Hu et al., 2023; Chen et al., 2023) addresses related threats, but no black-box attack (Xu et al., 2022; Wen et al., 2024) framework specifically targets HNNs, a gap this paper fills. For more details on these works, please refer to the Appendix C.

## 7. Conclusion

In this work, we explore black-box adversarial attacks on hypergraph-based models, revealing their structural vulnerabilities. We introduce HyperNear, the first node injection attack framework for HNNs, leveraging homophily constraints for stealth. Experiments demonstrate its high attack efficacy, transferability, and unnoticeability. Our analysis of unnoticeability metrics deepens the understanding of hypergraph structure and model robustness, paving the way for future work on strengthening hypergraph-based models.

## Acknowledgements

This work is supported by National Key Research and Development Program of China (2024YFE0214000), the National Natural Science Foundation of China (Grant Nos. U22A20102, 62337001, 62172370, T2122020), and the Jinhua Science and Technology Plan (No. 2023-3-003a).

## Impact Statement

This paper presents work whose goal is to advance the field of Machine Learning. There are many potential societal consequences of our work, none which we feel must be specifically highlighted here.

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

# A. Supplementary Materials

## A.1. Further Discussion

Hypergraphs are commonly used to model complex relationships between entities, often relying on homophily, the tendency for similar nodes to cluster together, as a key structural property that enhances predictive accuracy. In domains like social networks, recommender systems, and biological networks, this property forms a crucial foundation for tasks such as community detection and classification.

It is well understood that a higher homophily generally helps GNNs achieve better performance on downstream tasks, as neighbors positively influence the representation of the central node during aggregation (Zheng et al., 2022; Zhu et al., 2022; Luan et al., 2022; Huang et al., 2024). Reducing homophily between nodes disrupts the neighborhood aggregation process of graph-based model, resulting in performance degradation. However, this disruption can also be leveraged as an indicator of a potential attack. Therefore, by self-constraining the actions of attacker with respect to homophily, the attack can achieve greater stealth and evade detection more effectively. Also, the effect of other attack methods on the homophiliy ratio as shown in Figure 7.

Although this paper utilizes homophily as a restrictive metric for designing attack strategies, the complex structure of hypergraphs presents opportunities for deeper exploration. Investigating more sophisticated metrics for attack stealth will help uncover the fundamental principles needed to protect the stability of hypergraph models.

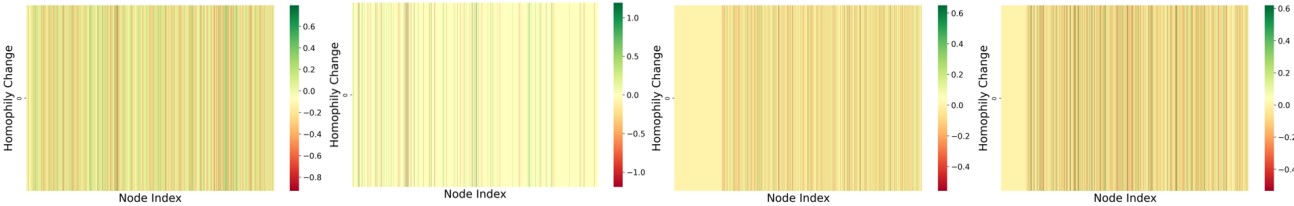

*Figure 7.* Impact of other attack methods on homophily ratio. The left two visuals show NDA attacks on the DBLP-CA dataset and Cora dataset, while the right two represent FGA attacks on the Pubmed dataset and Citeseer, respectively.

**Relationship Between Changes in FHH and Changes in Hypernode Feature.** Suppose that the hypergraph is perturbed by injecting nodes, and that the perturbation affects the set of neighbors $\mathcal{R}_v^t$ of node $v$. This perturbation causes a change in the node feature aggregation function, denoted as $f_t'(\cdot)$. By Taylor expansion, we know that the feature change $\Delta \mathbf{h}_v^{(k)}$ of supernode $v$ is denoted as:

$$\Delta \mathbf{h}_v^{(k)} = \sum_{t=0}^{n} \frac{\partial \phi}{\partial f_t} \Delta f_t + O\left((\Delta f_t)^2\right). \tag{15}$$

According to the definition of Eq. 10, FHH measures the similarity between the features of node $v$ and its neighboring feature aggregates. Then the change of FHH after perturbation can be approximated as:

$$\Delta FHH_v \approx \text{sim}(\mathbf{H}_v', \mathbf{X}_v') - \text{sim}(\mathbf{H}_v, \mathbf{X}_v). \tag{16}$$

Similarly, using the Taylor expansion, the result is obtained:

$$\Delta FHH_v \approx \sum_{t=0}^{n} \frac{\partial \phi}{\partial f_t} \Delta f_t + O\left((\Delta f_t)^2\right). \tag{17}$$

This suggests that changes in $FHH$ are directly caused by changes in the characteristics of node $v$ and its neighbors, which can be captured by the perturbation function $\Delta f_t$ and the partial derivatives of the corresponding aggregation function $\frac{\partial \phi}{\partial f_t}$.

We believe that promoting a shift from heterophily to homophily is key to developing an unnoticeable attack. Our empirical findings indicate a significant reduction in the homophily ratio of hypergraphs following adversarial attacks. Homophily, which measures the similarity between a node's features and those of its neighboring nodes, is essential for preserving the structural integrity of hypergraphs. When homophily is disrupted, the structural coherence of the hypergraph is compromised,

making the attack more noticeable and less effective. Therefore, maintaining homophily is crucial for ensuring that the attack remains unnoticeable and achieves its desired impact. In hypergraphs, where nodes are connected by hyperedges that capture complex high-order relationships, even small changes in homophily can lead to significant topological shifts due to the collaborative nature of the hypergraph structure. As shown in Eq.(17), small perturbations may cause nonlinear changes, especially in higher-order structures where the effects may be amplified as the perturbations propagate through multiple nodes.

By integrating homophily constraint into the overall loss function, we ensure that our adversarial perturbations are optimized for both attack effectiveness and unnoticeable. The full adversarial objective is expressed as:

$$\max_{||\mathcal{G}'-\mathcal{G}||\leq\mathcal{B}} \mathcal{L}_{atk} = \mathcal{L}_{cls} - \lambda\mathcal{L}_{FHH}, \tag{18}$$

where $\mathcal{L}_{cls}$ is the classification loss and $\lambda$ is a hyperparameter controlling the trade-off between classification performance and homophily preservation, and $||\mathcal{G}' - \mathcal{G}|| \leq \mathcal{B}$ ensures that the adversarial modifications stay within the allowed perturbation budget $\mathcal{B}$.

## A.2. Experimental Setups

In our study, we follow the strict black-box setup, which prohibits any querying of the model and only allows access to the incidence and feature matrices of the input data. The default hyperparameter settings in our method are as follows, the parameter random seed $s$ is 4202, the choice of hyperedge $k$ is 10%, the ratio of the injected nodes $\alpha$ is 5%, the effect of controlling the loss of homophily $\lambda$ and $\tau$ is 0.1. The comparison method is subjected to the same budgetary constraints. By default, the victim model and the target model are the same, where the victim model is the model used by the user and the victim model is the pre-trained model used by the agent model, which is the model attacked in our experiments. All experiments are conducted on a device with AMD EPYC 7543 32-core processor and a NVIDIA RTX A6000 GPU with 48 GB of RAM.

## A.3. Experimental Results

We provide visualization results on other datasets, as shown in Figure 8 and Figure 9.

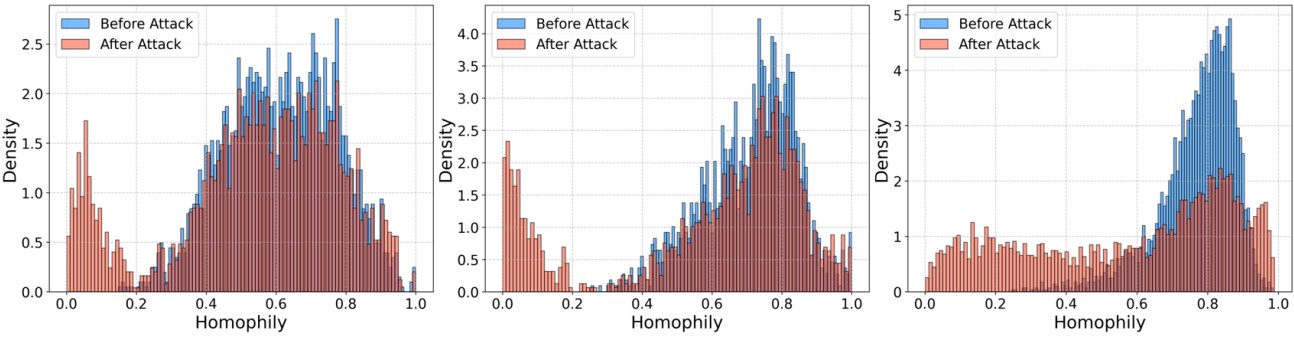

*Figure 8.* Changes in the homophily ratio. From left to right, Cora-CA dataset, Citeseer dataset, Pubmed dataset.

We also evaluated the portability of HyperNear to other HNN models in two additional datasets to complement the experimental results in the main text, as shown in Table 4.

## A.4. Datasets

We evaluate our models on five real-world hypergraph datasets for hypernode classfication tasks, including DBLP (Rossi & Ahmed, 2015), Pubmed, Citeseer and Cora (Sen et al., 2008). These are standard academic web datasets, where each node represents a document. For the DBLP and Cora datasets, a co-authorship hypergraph is constructed, with all documents co-authored by the same author forming a single hyperedge. In the case of PubMed, Citeseer, and Cora, a co-citation hypergraph is created, where each hyperedge links all documents cited by the same author. The statistics of datasets are provided in Table 5.

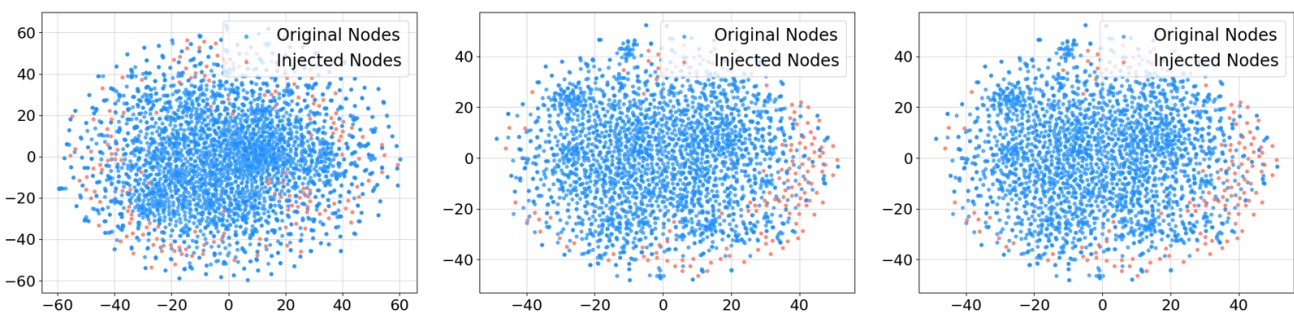

*Figure 9.* T-SNE visualization of data distribution after HyperNear attack. From left to right, Citeseer dataset, Cora dataset, Cora-CA dataset.

*Table 4.* Performance comparison of HyperNear on different HNNs.

| | CORA-CA | | CITESEER | |
| VICTIM MODEL | BEFORE | AFTER | BEFORE | AFTER |
| --- | --- | --- | --- | --- |
| HYPERGCN | $75.26 \pm 2.78$ | $57.36 \pm 7.42\downarrow$ | $70.76 \pm 0.92$ | $56.68 \pm 1.77\downarrow$ |
| HGNN | $82.10 \pm 0.86$ | $53.29 \pm 0.95\downarrow$ | $72.71 \pm 1.27$ | $46.14 \pm 1.53\downarrow$ |
| ED-HNN | $84.08 \pm 1.75$ | $62.38 \pm 1.18\downarrow$ | $73.45 \pm 1.36$ | $55.19 \pm 1.55\downarrow$ |
| UNIGCNII | $81.79 \pm 1.29$ | $70.28 \pm 1.42\downarrow$ | $73.26 \pm 1.16$ | $68.50 \pm 1.37\downarrow$ |

*Table 5.* Statistics of hypergraph datasets. The homophily ratio $\mathcal{H}$ is computed from Eq.(3) by labels.

| DATASETS | # NODES ($|\mathcal{V}|$) | # HYPEREDGES ($|\mathcal{E}|$) | # CLASSES | AVG.HYPEREDGE SIZE | LABEL RATE | LABLE-BASED HOMO. ($\mathcal{H}$) |
| --- | --- | --- | --- | --- | --- | --- |
| CORA-CA | 2,708 | 1,072 | 7 | 4.2±4.1 | 0.052 | 0.7797 |
| DBLP-CA | 43,413 | 22,535 | 6 | 4.7±6.1 | 0.040 | 0.8656 |
| CITESEER | 3,312 | 1,079 | 6 | 3.2±2.0 | 0.052 | 0.6814 |
| CORA | 2,708 | 1,579 | 7 | 3.0±1.1 | 0.052 | 0.7462 |
| PUBMED | 19,717 | 7,963 | 3 | 4.3±5.7 | 0.008 | 0.7765 |

## A.5. Notation Table

Table 6 summarizes the notations and definitions throughout this paper for clarity.

*Table 6.* Notation Table.

| NOTATION | DESCRIPTION | NOTATION | DESCRIPTION |
| --- | --- | --- | --- |
| $\mathcal{G}$ | A HYPERGRAPH | $\mathbf{x}$ | VECTOR OF VERTEX FEATURES |
| $\mathcal{V}$ | SET OF VERTEXS | $y$ | LABEL |
| $\mathbf{W}$ | DIAGONAL MATRIX OF HYPEREDGE WEIGHTS | $\mathcal{H}$ | LABEL-BASED HOMOPHILY RATIO |
| $\mathbf{X}$ | MATRIX OF VERTEX FEATURES | $\mathbf{I}_e$ | MATRIX BETWEEN THE ORIGINAL VERTEX AND THE NEW HYPEREDGE |
| $\mathbf{H}$ | INCIDENCE MATRIX | $\mathbf{I}_v$ | MATRIX BETWEEN THE NEW VERTEX AND THE ORIGINAL HYPEREDGE |
| $\mathbf{H}'$ | INCIDENCE MATRIX AFTER ATTACK | $\widehat{\mathbf{H}}$ | MATRIX BETWEEN THE NEW VERTEX AND THE NEW HYPEREDGE |
| $e$ | HYPEREDGE | $\mathcal{R}_v$ | SET OF ALL HYPEREDGES CONTAINING VERTEX $v$ |
| $v$ | VERTEX | $d_e$ | THE AVERAGE DEGREE OF HYPEREDGE |

## A.6. Algorithm

The implementation of our algorithm is summarized in Algorithm 1.

---

**Algorithm 1** HyperNear: Node InjEction Attack on HypeRgraph

---

1: **Input:** Node feature matrix $\mathbf{X} \in \mathbb{R}^{|\mathcal{V}| \times d}$, Hypergraph incidence matrix $\mathbf{H} \in \mathbb{R}^{|\mathcal{V}| \times |\mathcal{E}|}$, Injection ratio $\alpha$, Selected
    hyperedges $k$, Perturbation standard deviation $\eta$, Random seed $s$, Budget $\mathcal{B}$, Hyperparameter $\lambda$
2: **Output:** Modified node feature matrix $\mathbf{X}'$, Modified hypergraph incidence matrix $\mathcal{H}'$
3: Set random seed $s$
4: $n_{\text{injected}} \leftarrow \alpha \times |\mathcal{V}|$ {Number of injected nodes}
5: $\mathbf{X}_{\text{new}} \leftarrow$ Generate features for injected nodes using $\mathbf{X}$ with perturbation $\eta$
6: $\mathcal{E}_{\text{selected}} \leftarrow$ Select $k$ hyperedges from $\mathbf{H}$ randomly
7: **for** $i = 1$ to $k$ **do**
8:     Inject $m$ nodes from $\mathbf{X}_{\text{new}}$ into hyperedge $\mathcal{E}_{\text{selected}}[i]$
9:     Compute $\mathcal{L}_{cls}$ for the modified hypergraph
10:    Compute $\mathcal{L}_{FHH}$ based on feature homophily preservation
11:    **if** $||\mathcal{G}' - \mathcal{G}|| \leq \mathcal{B}$ **then**
12:       Calculate $\mathcal{L}_{atk} \leftarrow \mathcal{L}_{cls} - \lambda \mathcal{L}_{FHH}$
13:    **else**
14:       Revert the injection
15:    **end if**
16: **end for**
17: **while** adversarial objective not met **do**
18:    $\mathbf{H}_{update} \leftarrow$ Perform random split on the injected hyperedge in $\mathbf{H}$
19:    Compute $\mathcal{L}_{cls}$ for $\mathbf{H}_{update}$
20:    Compute $\mathcal{L}_{FHH}$ for the updated features
21:    $S_{inj} \leftarrow \arg\max_S \mathcal{L}_{cls}(f(\mathbf{H}_{update}), y)$ {Find the optimal splitting strategy}
22:    **if** adversarial loss increases or $||\mathcal{G}' - \mathcal{G}|| > \mathcal{B}$ **then**
23:       Reject the split
24:    **else**
25:       Accept the new hyperedge split $S_{inj}$
26:    **end if**
27: **end while**
28: **return** $\mathbf{X}'$, $\mathcal{H}'$

---

## B. Analyzing Hypergraph Topological Vulnerabilities

In this appendix, we present a detailed derivation of the theoretical framework that models the topological vulnerability of hypergraphs under adversarial perturbations, specifically focusing on hypergraph injection attacks. These attacks target hypergraphs by perturbing the node features and structure, causing significant changes in the model's performance. The impact of these perturbations is formally captured in Theorem 3.1. We begin by revisiting the general process of hypergraph convolution, before introducing the perturbations and analyzing their impact on the node feature vectors.

### B.1. Hypergraph Convolution Process

We start by defining the general hypergraph convolution process, which aggregates information from neighboring nodes in a hypergraph. The feature vector for node $v$ at the $k$-th layer, $\mathbf{h}_v^{(k)}$, is computed as a function of the feature vector from the previous layer, $\mathbf{h}_v^{(k-1)}$, and the aggregated features of the neighboring nodes:

$$\mathbf{h}_v^{(k)} = \phi(\mathbf{h}_v^{(k-1)}, f(\{\mathbf{h}_u^{(k-1)} \mid u \in e_j, e_j \in \mathcal{R}_v\})), \tag{19}$$

where $\mathcal{R}_v = \cup_{e_j \in \mathcal{E}} \{e_j \mid v \in e_j\}$ represents the set of hyperedges that contain node $v$, and $e_j$ (for $j = 1, 2, \ldots, |\mathcal{E}|$) denotes the $j$-th hyperedge. The functions $\phi(\cdot)$ and $f(\cdot)$ are vector-valued functions that update the node features based on the information from neighboring nodes.

This equation captures the process of updating the feature vector for node $v$ by aggregating the feature vectors of its neighbors across all hyperedges that contain $v$.

## B.2. Adversarial Injection Attack on Hypergraphs

In the context of adversarial attacks, we introduce the concept of a hypergraph injection attack. This attack targets specific areas of the hypergraph by injecting new nodes, perturbing the structure and feature vectors of the hypergraph without needing full access to the graph. The goal is to simulate the effect of small, targeted changes to the hypergraph that lead to significant changes in the model's output.

To model the effect of adversarial perturbations, we introduce a modified version of the aggregation function where the neighborhood sets are altered by injected nodes. Specifically, we define $\mathcal{R}_v^t$ as the set of $t$-hop neighbors of node $v$, and the corresponding aggregation function is denoted as $f_t(\cdot)$. The feature update process can then be rewritten as:

$$
\begin{aligned}
\mathbf{h}_v^{(k)} =& \phi\left(f_0(\{\mathbf{h}_v^{(k-1)}\}), f_1(\{\mathbf{h}_u^{(k-1)} \mid u \in e_j, e_j \in \mathcal{R}_v^1\}),\right.\\
& \left. f_2(\{\mathbf{h}_u^{(k-1)} \mid u \in e_j, e_j \in \mathcal{R}_v^2\}), \ldots, f_n(\{\mathbf{h}_u^{(k-1)} \mid u \in e_j, e_j \in \mathcal{R}_v^n\})\right).
\end{aligned}
\tag{20}
$$

In this formulation, the feature vector $\mathbf{h}_v^{(k)}$ is updated by aggregating the features from the node's $t$-hop neighbors across different layers of the hypergraph.

## B.3. Perturbation of the Hypergraph

Next, we introduce perturbations into the hypergraph. We assume that after the injection attack, the $t$-hop neighborhood of node $v$ changes from $\mathcal{R}_v^t$ to $\mathcal{R}_v'^t$, and the corresponding aggregation function $f_t(\cdot)$ changes to $f_t'(\cdot)$. The change in the feature vector $\mathbf{h}_v^{(k)}$ due to these perturbations can be approximated using a first-order Taylor expansion:

$$
\begin{aligned}
\mathbf{h}_v'^{(k)} =& \mathbf{h}_v^{(k)} + \frac{\partial \phi}{\partial f_0}(f_0' - f_0) + \frac{\partial \phi}{\partial f_1}(f_1' - f_1) + \cdots + \frac{\partial \phi}{\partial f_n}(f_n' - f_n) + O\left((\Delta f_0)^2 + (\Delta f_1)^2 + \cdots + (\Delta f_n)^2\right)\\
=& \mathbf{h}_v^{(k)} + \sum_{t=0}^{n} \frac{\partial \phi}{\partial f_t} \cdot (f_t' - f_t) + O\left((\Delta f_t)^2\right),
\end{aligned}
\tag{21}
$$

where $\Delta f_t = f_t' - f_t$ represents the perturbation in the aggregation function for the $t$-hop neighbors. The change in the feature vector $\Delta \mathbf{h}_v^{(k)}$ is then:

$$
\Delta \mathbf{h}_v^{(k)} = \mathbf{h}_v'^{(k)} - \mathbf{h}_v^{(k)} = \sum_{t=0}^{n} \underbrace{\frac{\partial \phi}{\partial f_t}}_{\text{Sensitivity of } \phi \text{ to } t\text{-hop features}} \cdot \underbrace{\Delta f_t}_{\text{Change in } t\text{-hop aggregated features}} + \underbrace{O((\Delta f_t)^2)}_{\text{Higher-order effects (nonlinear)}}.
\tag{22}
$$

This equation quantifies how the perturbations in the hypergraph propagate through the network and alter the feature vectors of the nodes.

## B.4. Weighted Aggregation and Perturbation Effects

For the case where the aggregation function uses a weighted average, the function $f_t(\cdot)$ can be written as:

$$
f_t(\{\mathbf{h}_u^{(k-1)} \mid u \in e_j, e_j \in \mathcal{R}_v\}) = \sum_{\{u \in e_j \mid e_j \in \mathcal{R}_v\}} w_u^t \mathbf{h}_u^{(k-1)},
\tag{23}
$$

where $w_u^t$ is the weight associated with the hyperedge between node $v$ and its $t$-hop neighbors. Under perturbations, the change in the aggregation function is given by:

$$
\Delta f_t = \sum_{\{u \in e_j \mid e_j \in \mathcal{R}_v\}} \left(\Delta w_u^t \mathbf{h}_u^{(k-1)} + w_u^t \Delta \mathbf{h}_u^{(k-1)}\right).
\tag{24}
$$

This expression demonstrates that the changes in both the structure (i.e., the weights) and the feature vectors of the neighbors amplify the perturbation effect.

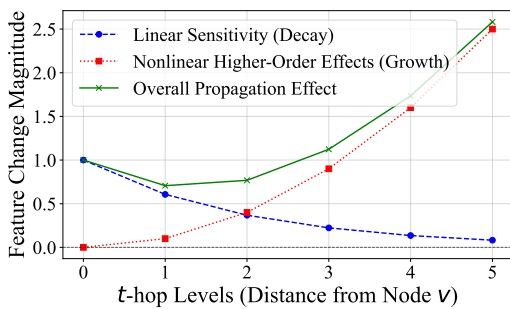

*Figure 10.* Illustration of perturbation propagation in HNNs. In Theorem 3.1, the first term models the primary contribution through linear propagation, which dominates for small perturbations. In contrast, the higher-order terms reflect nonlinear effects that, while smaller in magnitude, can become significant under large perturbations or high variability in hyperedge weights.

## B.5. Optimization Objective for Perturbations

A key aspect of adversarial perturbations is to minimize the magnitude of the change in the feature vector $\Delta \mathbf{h}_v^{(k)}$ while ensuring that the perturbations in the aggregation function exceed a certain threshold. This can be formulated as the following optimization problem:

$$\min_{\Delta \mathbf{h}_v^{(k)}} \parallel \Delta \mathbf{h}_v^{(k)} \parallel \quad \text{s.t.} \quad \sum_{t=0}^{n} \parallel \Delta f_t \parallel \geq \epsilon, \tag{25}$$

where $\parallel \Delta \mathbf{h}_v^{(k)} \parallel$ represents the magnitude of the change in the feature vector, and $\parallel \Delta f_t \parallel$ is the perturbation size in the aggregation function for the $t$-hop neighbors. The constraint ensures that the perturbation exceeds a predefined threshold $\epsilon$, which is necessary to produce significant changes in the output.

## B.6. Perturbation Propagation in HNNs

Figure 10 illustrates the sensitivity brought about by $t$-hop neighbors of node $v$ versus the change in uncertainty brought about by nonlinear higher-order effects on HNNs. These theorem and corollary offer a theoretical perspective on how perturbations in hypergraph topology propagate through the network. While the results illustrate the potential for small manipulations in hyperedges or node features to influence model behavior, the exact extent of this effect depends on specific model designs and datasets. As observed in Sec. 5.3, experimental results confirm that our injection attack framework, HyperNear, effectively exploits these vulnerabilities across multiple HNN models, demonstrating strong disruptive capabilities under diverse scenarios.

---

**Note: Optimization Objective for Adversarial Attacks**

To design effective injection attacks, the following optimization problem formalizes the trade-off:

$$\min_{\Delta \mathbf{h}_v^{(k)}} \parallel \Delta \mathbf{h}_v^{(k)} \parallel \quad \text{s.t.} \quad \sum_{t=0}^{n} \parallel \Delta f_t \parallel \geq \epsilon, \tag{26}$$

where $\parallel \Delta \mathbf{h}_v^{(k)} \parallel$ represents the magnitude of the change in the feature vector, and $\parallel \Delta f_t \parallel$ is a measure of the size of the perturbation in the aggregation function $f_t(\cdot)$, which we want to be at least $\epsilon$, a predefined threshold for significant change.

**Objective:** Identify minimal perturbations that lead to significant changes in the output.

---

## C. Related Works

**Adversarial Attack on Graph Data.** Existing attack methods (Dai et al., 2018; Zügner et al., 2020; Sun et al., 2022; Gosch et al., 2023) on graph data can be categorized into two types: graph modification attacks (Chen et al., 2018; Zügner et al., 2018; Zügner & Günnemann, 2019) and graph injection attacks (Sun et al., 2020; Zou et al., 2021; Ju et al., 2023; Zhang et al., 2024; Fang et al., 2024). Modification attacks achieve the desired attack effect by altering the graph structure, while injection attacks aim to disrupt the model's performance by injecting new nodes. Both types of attacks can significantly decrease the model's accuracy, however, injection attacks generally require less access privilege from the attacker. Thus, this paper focuses primarily on designing an injection attack framework.

Most prior work has only addressed simple graph structures (Wu et al., 2019), with edges representing direct connections between node pairs. In contrast, hypergraphs, which model higher-order relationships, have seen limited exploration in the context of adversarial attacks. This gap is particularly critical given the increasing adoption of HNNs in real-world applications, as the absence of robust understanding about potential vulnerabilities could expose these models to significant security risks.

**Hypergraph Neural Networks.** HNNs have been developed as an advanced extension of GNNs, specifically designed to capture high-order relationships that go beyond the pairwise interactions typical of traditional graph structures (Gao et al., 2020; Antelmi et al., 2023; Kim et al., 2024). Hypergraphs use hyperedges to simultaneously connect multiple nodes, enabling HNNs to model the more intricate and varied relationships found in real-world data. This feature makes HNNs particularly effective in domains such as social networks (Li et al., 2013; Yang et al., 2019), bioinformatics (Zheng et al., 2019; Shao et al., 2020; Xiao et al., 2019), and recommendation systems (Yu et al., 2021; La Gatta et al., 2022), where interactions often involve groups of nodes rather than individual pairs.

Various models have been proposed: HGNN (Feng et al., 2019; Gao et al., 2022), HyperConv (Bai et al., 2021), and HyperGCN (Yadati et al., 2019) process hypergraphs by converting them into traditional graphs through the hypergraph Laplacian operator. HyperMSG (Arya et al., 2020) leverages hypergraph structures by aggregating messages in a two-stage process. UniGNN (Huang & Yang, 2021) proposes a unified framework for both graph and hypergraph neural networks. HyperGCL (Wei et al., 2022) enhances the generalization ability of hypergraphs through contrastive learning, while ED-HNN (Wang et al., 2023) models high-order relationships using a hypergraph diffusion operator. HCoN (Wu et al., 2022) introduces hypergraph reconstruction error to train a classifier. These approaches enhance the ability of HNNs to handle large-scale data with complex multi-node relationships, enabling the development of hypergraph applications. Despite these advances, the security and robustness of HNNs remain underexplored.

**Remarks.** While some initial efforts (Hu et al., 2023; Chen et al., 2023) have explored related threats, these studies mainly focus on probing different data modeling approaches rather than designing attacks specifically for hypergraph models, which are central to many applications. Consequently, research addressing the unique structural characteristics of hypergraphs in adversarial scenarios remains scarce.

Our work directly tackles this gap by focusing on the structural vulnerabilities of HNNs under adversarial manipulation, with an emphasis on black-box attack settings. Black-box attacks (Xu et al., 2022; Wen et al., 2024), where adversaries generate adversarial samples without any knowledge of the target model's internal parameters or architecture, pose realistic and severe threats in practical scenarios. To the best of our knowledge, this is the first work to conduct adversarial attacks on hypergraphs in a black-box setting.

