# OpenReview forum: "HyperNear: Unnoticeable Node Injection Attacks on Hypergraph Neural Networks"
_ICML.cc/2025/Conference — ICML 2025 poster_

### Official Review · Reviewer_wtog · 2025-03-11

**Overall Recommendation:** 5

**Summary:**

This paper focuses on the vulnerability of hypergraph neural networks (HNNs) to node injection attacks. The authors introduce HyperNear, a novel node injection attack method specifically designed for HNNs, which exploits the homophily property to improve stealthiness. Through extensive experiments, the study demonstrates the effectiveness of HyperNear in black-box scenarios, representing a significant advancement in the field of adversarial attacks on hypergraphs. The findings underscore critical implications for the security and robustness of hypergraph-based models, offering valuable insights for future research in this area.

**Claims And Evidence:**

Yes, the claims made in the submission are supported by clear arguments and convincing results with detailed discussion.

**Essential References Not Discussed:**

The paper has covered the relevant related works necessary to understand its key contributions.

**Experimental Designs Or Analyses:**

The experiments are sound and substantiate the claims made in the theoretical analysis. The empirical results effectively showcase the impact of node injection attacks on the topology and homophily of hypergraphs.

**Methods And Evaluation Criteria:**

Yes.

**Other Comments Or Suggestions:**

1. Ensure that the main text references key sections of the supplementary material to guide readers through the additional content.
2. Ensure consistent font styles across all figures.
3. Increase the font size of the figures in the Experiments section.

**Other Strengths And Weaknesses:**

Strengths:
1. The paper provides a compelling motivation for investigating node injection attacks on hypergraphs, emphasizing the unique challenges introduced by higher-order dependencies. This is an interesting and important piece of work.
2. The paper is exceptionally well-organized, making it easy to follow. The motivation, methodology, theoretical analysis, and experiments are all presented in a coherent and logical manner.
3. The experimental design is comprehensive, and the open-source code facilitates further research in the field.

Weaknesses:
1. The comparison methods lack evaluation of stealthiness.
2. Where applicable, include summary rows or average performance rows to make it easier to compare the overall effectiveness of HyperNear against baselines.

**Questions For Authors:**

See the weaknesses and the above comments.

**Relation To Broader Scientific Literature:**

This paper fills a gap in research on adversarial attacks against HNNs. The authors effectively position their work within the broader literature by highlighting the unique challenges posed by higher-order dependencies in hypergraphs, which have been largely overlooked in prior research on adversarial attacks. The discussion of Finding 1 and Finding 2 provides a solid theoretical foundation that connects well with existing studies on graph-based adversarial attacks.

**Theoretical Claims:**

The paper provides a theoretical analysis on the impact of attacks on hypergraph topology and homophily, identifying two key findings: (1) Hypergraphs are highly sensitive to adversarial attacks due to their intricate interdependencies; (2) Naive attacks significantly disrupt homophily, which can be leveraged to design more subtle and effective attacks.

---

> ### Author Rebuttal · Authors · 2025-03-31
>
> We sincerely appreciate your **comprehensive and positive evaluation** of our work, particularly your recognition of our **theoretical contributions, experimental rigor, and positioning within the broader adversarial attack literature on hypergraphs**. Your feedback reinforces the significance of our study and motivates future research in this area. Next I address your concerns point by point.
>
> Weakness 1: In our paper, we have already evaluated the homophily shift for the ''Random'' baseline to illustrate stealthiness. Additionally, Figure 8 **provides a comprehensive analysis of how other attack methods (NDA, FGA) affect homophily across different datasets (DBLP-CA, Cora, Pubmed, Citeseer)**. This effectively demonstrates the stealthiness variations among different attack strategies. Your suggestion is valuable, and we will consider adding a table with homophily change metrics for other methods in the revised manuscript.
>
> Weakness 2: To improve the clarity of performance comparisons, we have added a summary row to the table that presents the average performance drop for each attack method relative to the ''Clean'' baseline. This allows for a more intuitive comparison of the overall impact of HyperNear against other methods.
>
> | Victim Model       | Clean | Random | NDA  | FGA  | HyperNEAR | Clean | Random | NDA  | FGA  | HyperNEAR | Clean | Random | NDA  | FGA  | HyperNEAR | Clean | Random | NDA  | FGA  | HyperNEAR |
> |--------------------|-------|--------|------|------|-----------|-------|--------|------|------|-----------|-------|--------|------|------|-----------|-------|--------|------|------|-----------|
> | Methods      |       |        |      |      |           |       |        |      |      |           |       |        |      |      |           |       |        |      |      |           |
> | UniSAGE          | Clean | Random | NDA  | FGA  | HyperNEAR | Clean | Random | NDA  | FGA  | HyperNEAR | Clean | Random | NDA  | FGA  | HyperNEAR | Clean | Random | NDA  | FGA  | HyperNEAR |
> | Avg. Drop ↓ | -     | 0.13   | 2.59 | 2.53 | 14.82     | -     | 0.2    | 2.75 | 2.93 | 13.97     | -     | -0.18  | 2.65 | 0.65 | 8.13      | -     | -0.48  | 2.67 | 1.01 | 8.53      |
>
>
> Suggestion 1: Thank you for the suggestion to improve cross-referencing between the main text and the supplementary material. We will scrutinize our manuscript againto **ensure that the reader is clearly directed to the relevant supplemental chapters** that provide more detail.
>
> Suggestion 2: Yes. We will ensure consistent font styles across all figures for a more polished presentation.
>
> Suggestion 3: We will increase the font size in the experimental figures to improve readability.
>
> Thank you again for recognizing this work and for your constructive comments.

---

> > ### Comment · Reviewer_wtog · 2025-04-04
> >
> > The authors have provided clear and satisfactory responses to the points I previously raised. I have also skimmed through their replies to the other reviewers, which reinforce my positive assessment regarding the originality and significance of the work. I have no remaining concerns and consider the paper ready for acceptance. Therefore, I am happy to update my score and recommend an SA.

---

> > > ### Author Response · Authors · 2025-04-09
> > >
> > > Thank you for acknowledging our clarifications and responses, as well as for your positive assessment of our work.
> > >
> > > Your constructive comments and suggestions have been instrumental in helping us further enhance the clarity and quality of our paper. We truly appreciate your time and support throughout the review process.

---

### Official Review · Reviewer_tPAy · 2025-03-11

**Overall Recommendation:** 4

**Summary:**

This work introduces HyperNear, a homophily-preserving node injection attack for hypergraph neural networks (HNNs). It provides a theoretical analysis of hypergraph vulnerability and demonstrates that homophily can be leveraged to enhance attack stealth. Extensive experiments show that HyperNear is highly effective and unnoticeable, making it the first black-box attack framework for HNNs, with important implications for hypergraph security.

**Claims And Evidence:**

The claims are well-supported by the presented evidence. The authors demonstrate that hypergraphs are vulnerable to node injection attacks through empirical analysis and propose a novel attack framework, HyperNear. The results show strong performance and generalization, validating their claims.

**Essential References Not Discussed:**

The paper provides a comprehensive overview of relevant works on HNNs and adversarial attacks on graph data. No critical gaps in the literature review were identified based on the provided context.

**Experimental Designs Or Analyses:**

The experimental design is well-structured, and the theoretical analysis is clearly presented and sound. I have carefully reviewed the relevant details and found them to be appropriate and convincing.

**Methods And Evaluation Criteria:**

The proposed methods are appropriate for the problem. The authors introduce a tailored attack framework for HNNs and evaluate it using extensive experiments in a black-box setting. The approach is sound and demonstrates effectiveness.

**Other Comments Or Suggestions:**

1.The font size in Table 1 could be increased to improve readability.

**Other Strengths And Weaknesses:**

Strengths
Novelty: The introduction of HyperNear as a node injection attack specifically designed for HNNs is a strong contribution.
Clear Motivation: The introduction effectively explains the research motivation. It contrasts traditional injection strategies with the paper’s core issue—achieving effective and stealthy attacks in hypergraphs. This sets a clear direction for the research.
Clear Mathematical Formulations: The paper rigorously defines its attack methodology and impact analysis.
Strong Empirical Results: The experiments show that HyperNear is effective across multiple datasets and models.
Weaknesses
1.The diagram in Figure 4 is somewhat unclear. It would be helpful to provide more context or explanation in the caption to clarify what the classification results represent and how they support the paper’s claims about the attack's effectiveness. Additionally, improving the visual clarity (e.g., clearer legends, or annotated highlights) could make the figure more informative.
2.Figures 1(a) and 1(b) could better illustrate how injected nodes integrate into the network through homophily, beyond just connections and attributes. This would highlight the role of homophily in stealthiness more clearly.
3.The authors have conducted experiments on five datasets (Cora-CA, DBLP-CA, Citeseer, Cora, and Pubmed), which effectively demonstrate the potential of HyperNear. To further strengthen their findings, it might be beneficial to explore a more diverse range of datasets. This would help ensure that the method remains effective across different scenarios.

**Questions For Authors:**

1.The paper states that hypergraphs' intricate dependencies can amplify the effects of small perturbations. Could you provide a more explicit quantitative measure or theoretical derivation to demonstrate this amplification effect?
2.I'm curious why the theoretical analysis in the paper focuses primarily on homophily. Is this the only or the best measure to evaluate the impact of hypergraph attacks?

**Relation To Broader Scientific Literature:**

The paper introduces a valuable contribution to adversarial attacks on hypergraphs. The study of vulnerabilities in graph neural networks is an active area of research. This paper extends this line of inquiry to hypergraphs, offering novel insights that are relevant to both theoretical and applied researchers.

**Theoretical Claims:**

Yes. I have checked carefully the proofs and details. Theorem 3.1 and Corollary 3.2 are well-grounded.

---

> ### Author Rebuttal · Authors · 2025-03-31
>
> We sincerely appreciate your detailed review and for recognizing the novelty and contributions of our work, including the introduction of HyperNear, its theoretical grounding, and its strong empirical results. Below, we address your comments point by point.
>
> Weakness1: Figure 4 illustrates the difference in the distribution of homophily ratio before and after the attack. We will refine Figure 4 by enhancing its clarity, improving the legend contrast, and adding explanatory annotations to highlight how classification results support our stealthiness claims. Thank you for your valuable suggestions.
>
> Weakness2: We clarify that Figure 1(a) is designed to illustrate **how the impact of injected malicious nodes propagates through the hypergraph topology**, rather than demonstrating homophily-guided injection. Meanwhile, we acknowledge that Figure 1(b) could better highlight homophily's role in stealthiness. To improve clarity, we will enhance the figure and captions to better convey these distinctions.
>
> Weakness3: Thank you for the suggestion. The five datasets we use are widely adopted benchmarks in hypergraph learning, making them appropriate for evaluating HyperNear. However, we agree that exploring additional datasets could further reveal the impact of hypergraph structure on attack robustness. We consider this an important direction for future work.
>
> Other 1: We will increase the font size in Table 1 to improve readability.
>
> Question 1: Our paper already provides a theoretical analysis of how hypergraph topology influences homophily and how local perturbations can cascade through the structure. Specifically, Theorem 3.1 formalizes how perturbations affect node relationships, **with the first term capturing linear propagation effects, while higher-order terms reflect nonlinear amplification**, which can become significant under large perturbations or varying hyperedge weights.
>
> Additionally, **Figure 11** illustrates how perturbations propagate in HNNs, visually demonstrating this cascading effect. The primary term dominates for small perturbations, ensuring controlled degradation, while higher-order dependencies introduce more complex, nontrivial effects under larger structural changes.
>
> Question 2: In hypergraphs, **homophily quantifies the similarity of connected nodes in feature space**, making it a natural and intuitive measure of structural changes. Since **hypergraph neural networks heavily rely on homophily for information propagation**, perturbing homophily can significantly degrade model performance, which aligns with our attack objectives.
>
> As discussed in our paper, **exploring alternative structural change metrics in hypergraphs is an important future research direction**. While homophily provides a strong foundation, other potential measures could offer complementary insights into structural vulnerabilities.
>
> Once again, we sincerely appreciate your thoughtful feedback and your recognition of the novelty and significance of this work. Your comments have helped us refine our presentation and identify valuable directions for future research.

---

### Official Review · Reviewer_FH69 · 2025-03-13

**Overall Recommendation:** 3

**Summary:**

This paper proposes a black-box node injection attack on Hypergraph Neural Networks (HNNs), named HyperNear. Unlike previous gradient-based white-box attacks on HNNs, this method does not require access to model parameters or gradients. Instead, it strategically injects malicious nodes and optimizes their connections to maintain high homophily, making the attack less detectable. The paper formulates the attack as an optimization problem, balancing attack effectiveness and stealthiness. Experimental results demonstrate that HyperNear significantly degrades classification performance while maintaining structural similarity to the original hypergraph, suggesting its robustness in real-world scenarios.

**Claims And Evidence:**

The paper claims that HyperNear effectively degrades HNN classification performance while remaining stealthy due to homophily-aware node injection. While the empirical results support the performance degradation claim, the assertion that homophily-based attacks are harder to detect lacks rigorous mathematical or experimental justification. Additionally, the paper does not discuss whether existing adversarial defense mechanisms can counteract this attack, leaving its robustness against defenses unclear.

**Essential References Not Discussed:**

None

**Experimental Designs Or Analyses:**

The experimental design follows standard practices, evaluating the attack on benchmark datasets with classification accuracy as the primary metric. The results demonstrate a clear performance drop, supporting the attack's effectiveness.

**Methods And Evaluation Criteria:**

The proposed method aligns with the problem setting, leveraging node injection to attack HNNs under a black-box assumption. The evaluation uses standard benchmark datasets, which are appropriate for assessing classification performance.

**Other Comments Or Suggestions:**

None

**Other Strengths And Weaknesses:**

Strengths:
S1. Novel black-box attack formulation for HNNs.

S2. Optimization framework balances attack effectiveness and stealthiness.

S3. Empirical evaluation on multiple benchmark datasets.

Weakness

W1: As described in the Related Works section, previous studies have already proposed gradient-based adversarial attack methods for HNNs. Therefore, the key focus of this work should be on transferring the attack to a black-box setting. I believe the novelty and motivation should emphasize black-box attacks rather than HNNs themselves.

W2: In Figure 1(b), I do not clearly see the distinction between Homo. and Hete.. Although I eventually realized that the authors might be trying to illustrate that injection attacks targeting Homo. structures are harder to detect, this conclusion requires strict mathematical or experimental support.

W3: The paper does not seem to discuss whether existing adversarial defense methods can detect or mitigate this attack.

**Questions For Authors:**

Q1: Compared to black-box attacks on GNNs and white-box attacks on HNNs, what are the key challenges of this paper’s attack on HNNs?

Q2: Can the attacker obtain complete information about the hypergraph? If they have full access to the graph structure and node attributes, would this still be considered a black-box attack?

**Relation To Broader Scientific Literature:**

The paper extends prior work on adversarial attacks against HNNs by shifting from gradient-based white-box methods to a black-box node injection attack, aligning with broader research on adversarial robustness in graph-based learning. It builds on the concept that homophily influences attack effectiveness, which has been explored in GNN adversarial studies but is less studied in hypergraph settings.

**Theoretical Claims:**

The paper provides a mathematical formulation for the attack optimization process, defining the objective function that balances attack effectiveness and stealthiness, though it lacks formal guarantees on its convergence and optimality.

---

> ### Author Rebuttal · Authors · 2025-03-30
>
> Claims&W2&W3: Thank you for your valuable feedback and for recognizing the effectiveness of our attack methodology.
>
> Our claim of stealthiness is based on the observation that homophily-aware attacks introduce perturbations aligned with existing structural patterns, making them less detectable than naive attacks. While **Figure 4 & 5 provide empirical evidence**, we acknowledge the need for stronger theoretical justification. Regarding adversarial defenses, existing graph-based methods may not directly apply to hypergraphs due to structural differences. Evaluating robustness against hypergraph-specific defenses is an important direction for future work.
>
> To the best of our knowledge, no prior work has established attack methods for hypergraphs. Our study serves as an exploratory step, proposing the first black-box attack framework for hypergraph neural networks, which is a fundamental prerequisite for systematic defense research. Given the absence of standardized adversarial defenses in hypergraph learning, we deliberately **prioritize vulnerability analysis over premature defense development**, aligning with security research paradigms in graph learning [4], where attack understanding precedes defense formulation. We believe **this work lays a crucial foundation for future robustness studies in hypergraph learning**.
>
> Theoretical: Thank you for recognizing our mathematical formulation of the attack optimization process, which balances effectiveness and stealthiness. While we do not provide formal convergence guarantees, this is due to the complexity of hypergraph structures, where high-order dependencies and discrete perturbations make theoretical analysis non-trivial. Nevertheless, **our empirical results consistently demonstrate stable and effective attack performance across multiple datasets**. Future work could explore theoretical analyses, such as proving convergence under specific constraints or employing continuous relaxations of the attack space.
>
> W1: While we agree that black-box attacks are a key contribution, our work is not merely about transferring existing attacks to a black-box setting. Instead, we position this as an exploratory study into the fundamental vulnerabilities of hypergraph neural networks under adversarial attacks, an area that has received little attention.
>
> Notably, every reviewer recognized our work as a promising and significant direction, **highlighting the importance of understanding adversarial threats in hypergraphs**. Our study emphasizes the challenges posed by the higher-order dependence of hypergraphs and the need for homophily-aware perturbations. We further elaborate on these challenges in Q1, where we discuss the difficulty of designing attacks that remain stealthy while coordinating complex hypergraph structures.
>
> W2: Thank you for the suggestion. We will enhance Figure 1(b) by improving color contrast, making the distinction between homophilous and heterophilous hyperedges more visually intuitive. For concerns regarding formal proofs, please refer to our response to Claims&W2&W3.
>
> Q1: Compared to black-box attacks on GNNs, perturbations in **standard graphs only affect pairwise connections, while in hypergraphs, a single hyperedge impacts multiple nodes**. This requires coordinated modifications to remain stealthy, making attack design significantly harder.
> Compared to white-box attacks on HNNs, **our black-box setting lacks access to model parameters or gradients**, requiring surrogate models and heuristic strategies.
>
> Moreover, hypergraph attacks require greater stealth, as their intricate interdependencies render naive perturbations more easily detectable. Additionally, while hypergraphs have attracted growing interest for their strong representational capabilities, the study of homophily in hypergraphs has only recently begun to receive attention [5]. The structural complexity of hypergraphs makes the homophily problem particularly challenging and still largely underexplored.
>
> Q2: We follow the strict black-box setting in [6], where attacker can access node features but has no knowledge of model parameters or gradients. Therefore, using original feature for feature generation is consistent with this definition. In addition, the use of raw features belongs to the reasonable a priori knowledge of the attacker (e.g., publicly available user features are observable by the attacker in social networks) and **does not conflict with the black-box setting**.
>
> Thank you again for recognizing this work and for your constructive comments.
>
> [4] Zügner et al., Adversarial Attacks on Neural Networks for Graph Data. In KDD, 2018.
>
> [5] Li et al., When Hypergraph Meets Heterophily: New Benchmark Datasets and Baseline. In AAAI, 2025.
>
> [6] Xu et al., Blindfolded Attackers Still Threatening: Strict Black-Box Adversarial Attacks on Graphs. In AAAI, 2022.

---

### Official Review · Reviewer_LMBD · 2025-03-14

**Overall Recommendation:** 1

**Summary:**

The authors proposed a node injection attack algorithm for hypergraph neural networks in black-box setting.

**Claims And Evidence:**

Problematic claims:
1. Un-noticability: I do not understand why the authors claim unnoticability where Figure 4(a) clearly distinguishes that "After attack" distribution is bimodal while "before attack" distr. is unimodal. Similarly, Figure 5(b) shows that one can identify the injected nodes in the periphery region.
2. Black-box claim. In the Feature Generation step of the algorithm, since the attacker uses the original node feature x_{ori} to construct the node feature of the injected node x_{inj}, could such an attack still be considered a black box?
3. Transferability: I am confused about the setting of the transferability experiment as it is not discussed properly. Have you considered any surrogate models other than HyperGCN, HGNN, ED-HNN and UniGCNII to generate perturbed Hypergraph $H'$?

**Essential References Not Discussed:**

None, to the best of my knowledge.

**Experimental Designs Or Analyses:**

The x-axis in Figure 4 => Does it indicate FHH, or node-label based homophily rate? I understand the reason to analyse Homophily distribution to justify un-noticeability, but a more reasonable way is to look at simpler statistics such as degree and dimension distribution.

**Methods And Evaluation Criteria:**

1. I understand the reason to analyse Homophily distribution to justify un-noticability, but a more straightforward and practical way is to look at simpler statistics such as degree and dimension distribution.

2. However, the baseline models are inadequately discussed in section 5.1. The authors should consider explaining Random, NDA and FGA more precisely in detail. For instance, it is not clear how "randomly generating node features" work. Or what kind of modification of features are considered for the smallest deg node in NDA?

**Other Comments Or Suggestions:**

Figure 2 is not good enough. Legends are obfuscating the numbers.

**Other Strengths And Weaknesses:**

The work is promising but needs more time to be publication-ready.

**Questions For Authors:**

1. Lines 25-27: “These attacks are
stealthy and practical, as they avoid altering existing nodes
or hyperedges. “ - why such attacks are practical in a hypergraph context?
2. Line 17 in Algorithm 1 says, “while adversarial objective not met do .. “. What are the conditions for the objectives to meet?
3. Table 2, Why some random injection attacks are invalid?
4. Theorem 3.1, What is the nature of the perturbation $\Delta \mathcal{R}_v$?
5. Equation 7, What do you mean by ``t-hop features’’? $f_1, f_2,...,f_n$ was never properly defined before? What are the choices and properties of $f()$ and $\phi$? Do you consider them differentiable? Please properly articulate the conditions under which the theorem holds.
6. Defn 3.4, What is the definition of average degree of hyperedge $e$? Please be clear and concise.
7. Proposition 3.3, “Naive adversarial attacks cause a significant
reduction in the homophily ratio of hypergraphs,..” - What constitutes a naive adversarial attack?

**Relation To Broader Scientific Literature:**

The broader direction of the paper is promising.

**Theoretical Claims:**

One of the key theoretical claims of the paper shows that the topology of a hypergraph is vulnerable to adversarial attack (sec 3.1). Apart from under-explained notations (articulated in ``Other comments’’), the most problematic part is that it is not clear how hypergraphs amplify minor perturbations in comparison to graphs. A more convincing argument is lacking here.

I did not find anything specific to hypergraph structure being shown as contributing to the Amplification effect. Similar arguments can be made for graphs with GNNs aggregation. For instance, lines 184-186 say, “A single perturbation can propagate through the structure, affecting multiple nodes simultaneously and highlighting the fragility of hypergraphs.”; the same thing can be said for a graph node that is connected to multiple other nodes.

---

> ### Author Rebuttal · Authors · 2025-03-30
>
> We sincerely appreciate your detailed feedback and address your concerns below.
>
> Claims1: Our claim of unnoticeability is **relative**, meaning that our attack is designed to be less detectable compared to naive perturbations. **The degree of unnoticeability also varies across datasets due to differences in hypergraph structures [1]**. For instance, Cora has a noisier and less structured hypergraph compared to Cora-CA, as their hyperedge constructions differ. Thus it produces less impact in Fig.4(b) (Cora-CA) than in Fig.4(a) (Cora).
>
> Claims2: Yes. Please refer to our response to Reviewer FH69’s Q2.
>
> Claims3: In addition to four surrogate models in transferability (HyperGCN, HGNN, ED-HNN, UniGCNII), Table 2 includes UniSAGE, UniGIN, UniGCN, and UniGAT, **covering diverse hypergraph neural networks** for comprehensive evaluation.
>
> Methods1&Exp: Our study focuses on homophily because both structural and feature perturbations directly affect it, making it a better indicator of unnoticeability than degree or feature distributions, which are outside our scope. As noted in line 201, we measure homophily FHH, defining the x-axis in Figure 4. Revealing homophily’s sensitivity is a key contribution, as it captures both structural and feature-based perturbation effects.
>
> Methods2: Random: Node features are randomly sampled from a Gaussian distribution fitted to existing node features. NDA: We select nodes with the smallest degrees and perturb their features by adding Gaussian noise proportional to feature variance. Full details are available in our public code.
>
> Theoretical1&Theoretical2: Section 3.1 analyzes hypergraph topology's vulnerability, not its comparison to standard graphs. Hyperedges amplify perturbations differently from graphs, where message passing is pairwise. While both graphs and hypergraphs allow perturbation propagation through the structure, **their structural mechanisms differ fundamentally**. Corollary 3.2 shows hyperedge weight and feature changes impact all incident nodes simultaneously.
>
> Supplementary: Thank you for noting this issue. The incomplete code was due to an outdated version of the repository used at the time of submission. We have **now updated it with the correct, runnable code**.
>
> Other: We will redraw Figure 2 to improve clarity, ensuring legends do not obscure numbers.
>
> Q1: The practicality of such attacks lies in the non-i.i.d. nature of hypergraph data, where structural dependencies significantly impact representation learning. Unlike graphs, hypergraphs propagate information across multiple entities, influencing representation learning (e.g., HGNN+ [2]).
>
> Q2: Lines 916-927 (Appendix B) define our adversarial objective. Algorithm 1 (Line 17) terminates when further perturbations **no longer improve attack effectiveness** or when **accuracy drops below a threshold**, preventing over-perturbation and also to conserve resources.
>
> Q3: We believe that, in some cases, random injections may unintentionally act as data enhancement, leading to performance improvements rather than degradation. This aligns with findings in contrastive learning for hypergraphs (e.g., HyperGCL [3]), where certain hyperedge augmentation can enhance model generalizability and robustness. This further justifies the need for dedicated adversarial attack studies on hypergraphs.
>
> Q4: Theorem 3.1's perturbation $ΔR_v$​ refers to **structural modifications via node injection**, altering neighborhood relationships and feature propagation.
>
> Q5: ''t-hop features" are neighborhood features aggregated through t-hop hyperedges. Line 110 defines $f(\cdot)$ as the aggregation function, with $f_t$ representing t-hop features. We will clarify that $n$ is the maximum hop count. As noted in Line 161, both $f(\cdot)$ and $\phi$ are differentiable hypergraph convolution aggregation functions. Theorem 3.1 holds if aggregation follows Eq. (6).
>
> Q6: Line 198 already provides the definition: the average degree of hyperedge $e$ is given by $d_e=\frac{1}{|e|}\sum_{i \in e}d_i$, ensuring balanced normalization in homophily calculations.
>
> Q7: Naive adversarial attacks lack strategic design and inject random or heuristic perturbations without considering hypergraph structure. Our optimized hypergraph-specific attacks balance effectiveness and stealth.
>
> We sincerely appreciate your thorough review and constructive feedback, which have greatly contributed to improving the clarity of our work. If you have any further questions or concerns, we would be glad to provide additional clarification or discussion. Thank you once again for your time and valuable input.
>
> [1] Wang et al., From Graphs to Hypergraphs: Hypergraph Projection and its Remediation. In ICLR, 2024.
>
> [2] Gao et al., HGNN+: General Hypergraph Neural Networks. In TPAMI, 2023.
>
> [3] Wei et al., Augmentations in Hypergraph Contrastive Learning: Fabricated and Generative. In NeurIPS, 2022.

---

> > ### Comment · Reviewer_LMBD · 2025-04-03
> >
> > Thank you for your effort in the response. I keep my score due to concerns below:
> >
> > Concerns about  -
> >
> > > Claim1:
> >
> > The paper claims to be the “first work to perform a global adversarial attack on HNNs in a black-box setting”.  The baselines are adaptation of attacks that were designed for graphs; not designed for hypergraphs. With such state of affairs, does it make sense to argue that the paper claims un-noticability in relative terms? See for instance, Mettack and Nettack for how to explicitly incorporate unnoticability criteria into attack algorithms.
> >
> >
> > > Claim3:
> >
> > Lines 365- 368 do not mention that you tested transferability on these models: UniSAGE, UniGIN, UniGCN, and UniGAT. This is confusing. When you discuss baseline method FGA on line 302, you mention about a “hypergraph proxy model” - What kind of proxy model is this? Please clarify this.
> >
> > > Methods1&Exp:
> >
> > “Our study focuses on homophily because both structural and feature perturbations directly affect it, making it a better indicator of unnoticeability than degree or feature distributions” => Do you have any theoretical or empirical justification for this statement that homophily is a better indicator of unnoticability than degree or feature distributions?
> >
> > > Method2:
> >
> > Thanks for clarifying.
> >
> > > Theoretical1&Theoretical2:
> >
> > “Section 3.1 analyzes hypergraph topology's vulnerability, not its comparison to standard graphs. “ => Why one needs to perturb hypergraph directly? Why can’t we transform it into a bipartite graph (or clique graph) and do our perturbations in bipartite graph space (or graph space)? This is why the vulnerability comparison in relation to such constructions are important.
> >
> > “Hyperedges amplify perturbations differently from graphs, where message passing is pairwise.” => Section 3.1 does not show how hyperedges amplify perturbations differently from equivalent graph-based constructs such as  clique graph or bipartite graph.
> >
> > “While both graphs and hypergraphs allow perturbation propagation through the structure, their structural mechanisms differ fundamentally.” => Not entirely true. You can transform a hypergraph to a bipartite graph, and doing so the perturbation propagation would not be any different.
> >
> > **Novelty concerns**
> >
> > The impact of homophily in Graph attack was already investigated in [3] for GNNs. The definition of homophily for hyperedges (Eqn 3) you defined appears to be a straightforward adaption:  homophily ratio[3] of the clique representation of a hyperedge.
> >
> > **Additional Suggestions:**
> >
> > 1. Please investigate how the performance compares with the baselines in evasion setting.
> > 2. Please conduct efficiency studies reporting the execution time of algorithm 1, along with a runtime complexity of Algorithm 1. It is recommended to have an efficiency subsection for the attack as done in Mettack [1]
> > 3. Please consider large-scale hypergraph datasets proposed recently such as [2] to evaluate if the attacks scales to large hypergraphs.
> >
> > Refs:
> >
> > [1] Adversarial Attacks on Graph Neural Networks via Meta Learning, ICLR’19
> > [2] Datasets, tasks, and training methods for large-scale hypergraph learning, 2023.
> > [3] How does Heterophily Impact the Robustness of Graph Neural Networks? Theoretical Connections and Practical Implications. KDD’22.

---

> > > ### Author Response · Authors · 2025-04-05
> > >
> > > >Claim1
> > >
> > > Our contribution lies not only in proposing a black-box attack setting, but also in **designing a principled, homophily-aware strategy specifically tailored to hypergraphs**. While Metattack is a gradient-based poisoning attack on graphs via meta-optimization, directly adapting it to hypergraphs fails to preserve high-order semantics unique to hyperedges. Our baseline methods are thus adapted for the hypergraph domain under the same setting. Our claim of unnoticeability is explicitly relative, i.e., our method achieves higher stealth compared to these adapted graph baselines on the same HNN tasks.
> > >
> > > >Claim3
> > >
> > > As we clarified in our previous response, the results of UniSAGE, UniGIN, UniGCN, and UniGAT are already presented in **Table 2**, demonstrating that our proposed method generalizes across a broad spectrum of hypergraph neural networks.
> > > The models added in **Lines 365–368** (HyperGCN, HGNN, ED-HNN, UniGCNII) were included to further enhance transferability coverage, not as exclusive targets.
> > >
> > > We also clarify that in black-box poisoning scenarios, a “surrogate model” refers to any model used by the attacker to approximate the victim’s behavior. This standard terminology may have led to confusion but is widely used in adversarial literature.
> > >
> > > >Methods1&Exp
> > >
> > > We chose homophily as a stealthiness indicator due to its theoretical and our empirical relevance. It reflects feature consistency within hyperedges and is influenced by both structural and feature perturbations. Alternative metrics such as degree shifts showed no consistent correlation with attack effectiveness, whereas **changes in homophily aligned well with model degradation across datasets**.
> > >
> > > We agree that combining multiple stealthiness metrics could further enhance attack characterization, and plan to investigate other topology-aware indicators as complementary tools in future work.
> > >
> > > >Concerns on Theoretical Results
> > >
> > > We appreciate the reviewer’s continued engagement. However, we would like to clarify a common misconception regarding the equivalence between hypergraphs and their graph-based projections (e.g., bipartite or clique graphs).
> > > While such transformations are mathematically feasible, **they fail to preserve the original structural semantics of hypergraphs, especially when it comes to high-order interactions [1] [7] [8]**.
> > >
> > > In Section 3.1, we therefore analyze vulnerability directly in the native hypergraph topology, not due to oversight, but precisely because **any analysis done in projected graph space would be fundamentally insufficient or misleading**. Our corollaries show how hyperedge perturbations jointly affect all incident nodes, a behavior unique to hypergraphs.
> > >
> > > In short, the question “Why perturb hypergraphs directly?” has been extensively studied and answered by the field. We regret if this foundational distinction was not clearly communicated, though we respectfully point out that our first-round response already cited this exact work (Ref [1]: Wang et al., ICLR 2024) to support our theoretical design and claims.
> > >
> > > >Novelty concerns
> > >
> > > We respectfully disagree with the claim that our homophily formulation and investigation lack novelty.
> > > First, the impact of homophily in graph attacks, as studied in [3], is explicitly acknowledged and cited in our main paper (see Line 215). Our work does not overlook prior contributions; instead, **it builds upon them and extends the homophily analysis to hypergraphs, which is a non-trivial and timely advancement**.
> > >
> > > Our homophily formulation (Eq. 3) is not an adaptation from clique expansions, which oversimplify hyperedges. Instead, it captures native hypergraph label cohesion without transformation, aligning with recent findings [7] that highlight the need for hypergraph-specific metrics. This further confirms that **the problem we study is not a marginal tweak on known graph results but rather part of a growing and distinct research frontier**.
> > >
> > > We also note that this perspective on novelty and homophily analysis has been positively acknowledged by other reviewers, further reinforcing the relevance and timeliness of our contribution.
> > >
> > > >Additional Suggestions
> > >
> > > Thank you for your suggestions. We agree that exploring efficiency and scalability is important. Due to space limits, our current submission focuses on methodological contributions and validating the proposed black-box threat model. We are extending our framework to larger-scale datasets and plan to report efficiency analyses in future work.
> > >
> > > Nonetheless, these points do not affect the core validity of our current findings.
> > >
> > > >Additional References
> > >
> > > [7]Li et al., From Heterophilous Graph Learning to Heterophilous Hypergraph Learning: Exploring New Frontiers. In Technical Report at IMS-NTU joint workshop on Applied Geometry for Data Sciences Part I, 2024.
> > >
> > > [8] Millán et al., Topology shapes dynamics of higher-order networks. Nat. Phys. 21, 353–361 (2025).

---

### Decision · Program_Chairs · 2025-05-01

**Decision:**

Accept (poster)

**Comment:**

This paper studies the vulnerability of hypergraph neural networks (HNNs) to node injection attacks and proposes HyperNear, a node injection attack method specifically designed for HNNs, which exploits the homophily property to improve stealthiness. Four reviewers reviewed this paper but with diverse scores. After rebuttal and discussion, three reviewers are positive, and one reviewer is still negative. The reviewer who increased the score from negative to positive still has concerns about the theoretical or empirical explanation supporting why unnoticeability is particularly crucial for hypergraph attacks. Based on the above facts, the AC would treat this paper as a borderline paper. Considering this paper is claimed to be the first one to explore the adversarial attacks on hypergraph neural networks (HNNs), the AC leans acceptance. The reviewers' comments and the rebuttal are suggested to be taken into account in the final version if this paper is accepted.